# Hailstorm Events in the Central Andes of Peru: Insights from Historical Data and Radar Microphysics

Jairo M. Valdivia[1,2], José Luis Flores-Rojas[1], Joseph J. Prado[4], David Guizado[1,3], Elver Villalobos-Puma[3], Stephany Callañaupa[3], and Yamina Silva-Vidal[1]

[1]Instituto Geofísico del Perú (IGP), Lima, Peru
[2]University of Colorado Boulder, Colorado, USA
[3]Instituto Nacional de Investigación en Glaciares y Ecosistemas de Montaña (INAIGEM), Lima, Perú
[4]Servicio Nacional de Meteorología e Hidrología del Perú, Lima, Perú

**Correspondence:** Jairo M. Valdivia (valdiviaprado.ing@gmail.com)

**Abstract.** Hailstorms, while fascinating from a meteorological perspective, pose significant risks to communities, agriculture, and infrastructure. In regions such as the Central Andes of Peru, the characteristics and frequency of these extreme weather events remain largely uncharted. This study fills this gap by investigating the historical frequency and vertical structure of hailstorms in this region. We analyzed historical hailstorm records dating back to 1958 alongside four years of observations (2017-2021) from Parsivel2 disdrometer and a cloud profiler radar MIRA35c. Our findings indicate a trend of decreasing hail frequency (-0.5 events/decade). However, the p-value of 0.07 suggests the need for further investigation, particularly in relation to environmental changes and reporting methods. The results show that hailstorms predominantly occur during the austral summer months, with peak frequency in December, and are most common during the afternoon and early evening hours. The analysis of radar variables such as reflectivity, radial velocity, spectral width, and linear depolarization ratio (LDR) reveals distinct vertical profiles for hail events. Two case studies highlight the diversity in the radar measurements of hailstorms, underscoring the complexity of accurate hail detection. This study suggests the necessity for refining the Parsivel2 algorithm and further understanding its classification of hydrometeors. Additionally, the limitations of conventional radar variables for hail detection are discussed, recommending the use of LDR and Doppler spectrum analysis for future research. Our findings lay the groundwork for the development of more efficient hail detection algorithms and improved understanding of hailstorms in the Central Andes of Peru.

## 1 Introduction

Hail is a type of precipitation that forms when strong thunderstorms produce updrafts that carry raindrops high into the atmosphere (Knight, 1981; Knight and Knight, 2001). As the raindrops are lifted higher and higher, they encounter freezing temperatures and become coated with ice. Eventually, these ice-coated raindrops grow to a large enough size and fall to the ground as hail (Knight, 1981). The formation of hail is a complex process that involves several microphysical processes (Pruppacher and Klett, 2010). Storms can produce hailstones from either type of embryo, including heavily rimed snow particles and frozen drops (Knight, 1981). Hailstones grow mainly by collecting supercooled liquid cloud droplets and raindrops within

the updrafts of convective storms (Pruppacher and Klett, 2010). These growth processes are further characterized by different growth regimes that describe the physical properties of the resulting hailstones (Ludlam, 1958; Lamb and Verlinde, 2011).

Typically, hailstones enter warmer air and partially or completely melt as they fall (Rasmussen and Heymsfield, 1987). Hail is defined as ice pellets that exceed 5 mm in diameter. When they reach the surface, hailstones exceeding 2 cm in diameter are considered severe (Raupach et al., 2021). However, even hailstones smaller than 2 cm can cause significant damage to crops.

Hailstorms pose a significant threat globally (Allen et al., 2020), leading to extensive damage and substantial financial losses (Púcik et al., 2019; Diaz and Murnane, 2008). In the USA, for instance, these storms result in losses of around US$10 billion every year (Diaz and Murnane, 2008), with a single severe hail event capable of causing over US$1 billion in damage (Changnon, 2009; Allen et al., 2017). Areas with high population density are particularly vulnerable, as demonstrated by the 1999 hailstorm in Sydney, Australia, the 2012 hailstorm in Phoenix, Arizona, USA, and the 2013 hailstorms over central and southwest Germany, which caused losses of billions of dollars (Kunz et al., 2018; Allen and Allen, 2016; Allen et al., 2017). Similarly, hailstorms in the Mantaro river valley in Peru, despite being located in a different region, can cause significant damage to crops, property, and infrastructure in the area, especially during the summer season when hailstorms are more prevalent.

In the Mantaro river valley and a large portion of land is owned by small and medium-sized producers who engage in crop diversification as a protective measure against extreme weather and market fluctuations, and climate change (Giráldez et al., 2020, and references therein). Over the years there have been changes in precipitation regimes with a reduction in precipitation trends (-5.6 mm/decade) in the last decade (Giráldez et al., 2020; Silva et al., 2008). Furthermore, anthropogenic climate change is expected to have significant impacts on hailstorms (Raupach et al., 2021), which can result in damaging consequences (Trapp et al., 2019, 2007; Brooks, 2013). The low-level moisture and convective instability (Diffenbaugh et al., 2013; Hoogewind et al., 2017; Rasmussen et al., 2020), melting level height (Dessens et al., 2015; Xie et al., 2008; Prein and Heymsfield, 2020), and vertical wind shear (Weisman and Klemp, 1984; Dennis and Kumjian, 2017) are all expected to change with geographical variability, leading to changes in hailstorm frequency and severity (Mahoney et al., 2012; Brimelow et al., 2017; Kunz et al., 2009). However, due to limited direct observations of hail, incomplete understanding of the microphysical and dynamical processes, and challenges in running models at sufficient resolution, the response of hailstorms to warming remains highly uncertain (Raupach et al., 2021; Allen et al., 2020). Despite its potential for damage, little is known about the frequency of hail occurrences in the region. In South America, there are few historical records of hailstorms, and there is no clear general trend in the frequency of these events. Most of these observations are from Argentina, which is one of the regions with the highest likelihood of hailstorms in the region (Prein and Holland, 2018). However, there are regional differences in the trend of hail occurrence. In northern Argentina, there is a positive trend in the frequency of hailstorms, while in the central and eastern regions of Argentina, a negative trend has been observed (Mezher et al., 2012). Beal et al. (2020) observed that on the southern border of Brazil and Argentina there are both positive and negative trends and stations that did not register significant trends. Studies on the effects of climate change on hail are rare in South America (Martins et al., 2017; Beal et al., 2020; Prieto et al., 1999) and there are no references in the literature on future projections of hail or storms over the region.

The global picture of how hailstorms will be affected by climate change in the future remains uncertain, with many regions unstudied, a lack of long-term observational data, gaps in understanding at the process level complicated by interactions between atmospheric variables relevant to hail, and limited modeling of hail. Consequently, Raupach et al. (2021) made five recommendations for future studies:

- Improved observational records of hail are required, including long and homogeneous data sets to enable separation of natural climate variability from potential trends due to anthropogenic climate change. When possible, observations should include both hail frequency and hailstone size.

- Statistical proxy relationships between environmental conditions and hail occurrence must be evaluated and improved. Proxy studies are extremely valuable, but current relationships typically cannot account for hailstorm initiation. These statistical links need to be tested in hail-resolving model simulations to detect possible future changes. Alternatively, machine learning approaches that allow for non-stationarity could be considered.

- Process-oriented studies are necessary, including detailed analysis of the microphysical chain of events leading to hail production within the storm and its final impact on the ground. Connections between dynamic processes on the synoptic and climate scales must also be understood. Investigations should focus on low-level moisture and convective instability, microphysics, and storm initiation. Field experiments and simulations can be used to enhance the understanding of fundamental microphysical processes.

- Changes to hail damage and its economic impact require deeper investigation, with attention not only on hail properties but also on changes to hail exposure and vulnerability. Possible investigation approaches include coupling convection-resolving simulations to impact models and projections of future population growth, and the use of statistical tools that can deal with non-stationarities in relationships between hazard, exposure, vulnerability and climate change. Future studies should also include modelling of the potential future economic impact of hail.

- More high-resolution numerical model simulations are required to better resolve hailstorm processes and to investigate expected future changes around the globe. As computational power increases, it will be possible to run convection-resolving simulations over larger regions and for longer time periods at increasing resolution and in ensemble modes.

In the Andes, apart from studies in the Argentine Andes (Prieto et al., 1999), to date, there are no studies on the climatology or frequency of hailstorms. In particular, the lack of quality meteorological data in the Andes has been a major limitation in understanding extreme weather events, such as hail. In this context, the Geophysical Institute of Peru (IGP) has played a fundamental role through the Huancayo Observatory. Aware of the need for quality meteorological data in the region, the IGP has established a set of instruments and sensors at the Huancayo Observatory, in the Laboratory of Atmospheric Physics and Microphysics and Radiation (LAMAR; Flores-Rojas et al. 2021b), to study the properties of the atmosphere and clouds, including hail.

In this article, we will explore the climatology and microphysics of hail in the Peruvian Andes, examining the observational features of its formation and the ways in which it can be studied and identified. To better understand the microphysics of hail

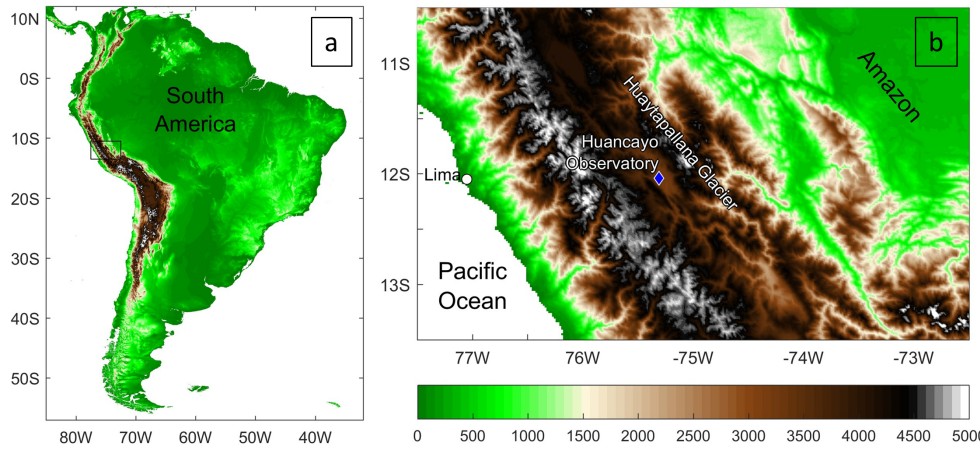

**Figure 1.** Location and topography surrounding the Huancayo Observatory. (a) Overview of the study area from a South American perspective. (b) Close-up view of the region outlined by the black box in (a). The Huaytapallana glacier is positioned to the east of the Huancayo Observatory. The color bar denotes altitude in meters above mean sea level (MSL).

and its frequency in the region, we will analyze historical reports of hailstorms dating back to 1958, as well as four years of observations using data from LAMAR. We will use the Parsivel2 disdrometer and a cloud profiler radar to identify hail events and focus on analyzing microphysical parameters associated with hail. The paper is organized as follows: Section 2 will describe the data and methodology, including the historical report of hailstorms and instrument characteristics. In Section 3, we will describe the results. The hail climatology and instrumental observations of hail will be described, and a couple of study

cases will be analyzed in detail. The implications and limitations of our findings will be discussed in Section 4, and finally, Section 5 will summarize the main findings and conclusions.

## 2  Data and Methodology

This study utilized data collected at IGP's Huancayo Observatory (12°01'18"S, 75°39'22"W), located at 3330 m MSL in the central part of the Peruvian Andes. The observatory houses LAMAR, a collection of instruments for atmospheric studies

(Flores-Rojas et al., 2021b), as well as a conventional weather station that has been in operation since 1922. LAMAR is within the Mantaro valley, which is part of the Mantaro river basin covering some territories of the Ayacucho, Huancavelica, Pasco, and Junin regions. The basin has an average altitude of 3870 m ASL, with an estimated annual precipitation of less than 1600 mm year$^{-1}$. Almost 83% of the annual rainfall occurs during the rainy season, which lasts from September to March. The rest of the rainfall occurs during the dry season, which lasts from April to August.

Historical hailstorm reports were collected and digitized for analysis. The information includes the date, time, and in some cases, the duration of the events, and is recorded daily by the meteorological observer. Data collection was done by Valdivia and Silva (2016) for the period of 1958 to 2016. Unfortunately, hailstorms stopped being reported in early 2016 due to staff

changes. The observer only reported hailstorms that were observed over the Huancayo Observatory, which are typically very localized. However, it is crucial to acknowledge that these historical reports are likely inclusive of both hail and graupel—two forms of solid precipitation that can be easily confused due to their similar appearance, particularly in the context of ground-based observations. Given that the observers recorded instances of solid precipitation they visually identified as hail, it is reasonable to surmise that some of the events cataloged might have actually been graupel, a softer and less dense form of frozen precipitation.

However, the instruments with advanced technology, such as the optical disdrometer Parsivel2 and the vertically pointing radar MIRA-35c, have been installed at the Huancayo Observatory to provide more accurate and comprehensive measurements of hydrometeors, including hail.

## 2.1 Optical disdrometer Parsivel2

Parsivel2 is the next version of the Parsivel, an laser-optical disdrometer that measure the size and velocity of hydrometeors (Löffler-Mang and Joss, 2000). The Parsivel2 has been evaluated through a comparative study at NASA's Goddard Space Flight Center (Tokay et al., 2014). The Parsivel2 sampling output is 1 min. The raw data provides the number of drops in a $32 \times 32$ size versus velocity matrix. The size range is from 0 to 25 mm, and the class width increases from 0.125 mm in small sizes to 3 mm in biggest sizes. The fall velocity range is from 0 to 25 m s$^{-1}$, and the class width increase with the fall velocity. The Parsivel2 has been operating since 2017. The internal software computes the rainfall parameters such as rain intensity, number of detected particles, particle velocity, and weather code. The Parsivel2 assigns a precipitation classification whenever precipitation is observed. The classification is based on the format of precipitation type SYNOP wawa4860 weather code developed by World Meteorological Organization (WMO 2019). The categories are: Drizzle, drizzle with rain, rain, rain drizzle with snow, snow, snow grains, freezing rain, hail (OTT, 2016). Parsivel2's internal processing filters out very small droplets (less than 0.7 mm) that are well above the theoretical droplet velocity. We are not applying any additional post-processing or filtering as our intention in the first instance is to evaluate the entire raw data matrix.

The Parsivel2 disdrometer has several limitations, as evidenced by the findings from various studies. One major limitation is the instrument's sensitivity, which may not be adequate for detecting very small hydrometeors, resulting in underestimation of the total number of particles and their size distribution (Battaglia et al., 2010; Park et al., 2017). Furthermore, the Parsivel2 can struggle to accurately classify irregularly shaped hailstones or complex snowflake shapes, leading to errors in size distribution measurements (Battaglia et al., 2010). The Parsivel2 measures rainfall at a single point, which might not be representative of the overall precipitation in the surrounding area, particularly during convective events with high spatial and temporal variability (Jaffrain et al., 2011; Jaffrain and Berne, 2011, 2012; Tokay et al., 2016; Angulo-Martínez et al., 2018; Jia et al., 2019). Environmental factors, such as wind, temperature, and humidity, can also impact the instrument's performance and the accuracy of its measurements (Friedrich et al., 2013a, b; Park et al., 2017). Several studies have attempted to correct the Parsivel2's measurements to improve their accuracy, such as adjusting the fall velocity of particles (Battaglia et al., 2010; Raupach and Berne, 2015) or investigating the effects of various factors on particle fall velocity (Jia et al., 2019).

**Table 1.** MIRA35c specifications.

| | |
|---|---|
| Frequency | 34.85 GHz |
| Peak power | 2.5 kW |
| Receiver | Single Polarization |
| Operation mode | Pulsed |
| Beam width | 0.6º |
| Antenna type | Cassegrain |
| Range resolution | 31 m |
| Temporal resolution | 5.6 s |
| Number of range gates | 415 |
| No. of spectral bins | 128 |

Under non-extreme conditions, the Parsivel2 disdrometer demonstrates reliable performance, especially in moderate rainfall ($R<10$ mm h$^{-1}$) and for midsize drops (0.6 - 4 mm in diameter) (Tokay et al., 2014; Park et al., 2017). At the Huancayo Observatory, comparisons between the Parsivel2 and rain gauges indicate a systematic underestimation of total rainfall (Valdivia et al., 2020b; Kumar et al., 2020; Del Castillo-Velarde et al., 2021). This underestimation bias is around of 16% for a single
rainfall event.

## 2.2 Vertically pointing radar MIRA-35c

The MIRA-35c is a high-frequency cloud profiler Ka-band radar. And it is sensitive to clouds and precipitation. MIRA-35c was manufactured by Meteorologische Messtechnik GmbH (METEK) and operates at a frequency of 34.85 GHz ($\lambda = 8.6 \ mm$), it uses a magnetron providing 2.5 kW pulse power and transmits a linear polarized signal, while two polarized signals are received
simultaneously. The system covers a range between 150 m and 13 km above the ground at a vertical range resolution of 31 m. Data from the range bins below 250 m are not used because of near-field clutter. The Table 1 shows the complete specifications of this MIRA35c installed at LAMAR. In this work the original software products are used in order to standardize the data. The analysis focuses on the 4 main radar variables, which are:

Reflectivity (dBZe): The equivalent radar reflectivity factor is a measure of the power returned to the radar, which is directly
related to the size and number of hydrometeors (raindrops, snowflakes, hailstones, etc.) in the radar's path. High reflectivity indicates the presence of large or numerous hydrometeors, which often means heavy precipitation. In the context of hail, high reflectivity near the surface could indicate the falling of hailstones.

Radial Velocity: This measures the speed at which the precipitation is moving toward or away from the radar. Since the radar is vertically pointing, the radial velocity is a measure of hydrometeor fall velocity plus the the wind vertical motion. In storms,
this can provide information about wind patterns and the movement of the storm itself. High radial velocities may indicate intense updrafts and downdrafts, which are associated with severe weather conditions such as hailstorms.

Spectral Width (RMS): This is a measure of the variability of the radial velocity within a single radar resolution volume. High spectral width values can indicate turbulence or it can be a measure of how different are the hydrometeors velocity or a combination of both. In severe weather situations like hailstorms, this turbulence can be caused by strong updrafts and downdrafts or by the collision and interaction of different types of hydrometeors.

Linear Depolarization Ratio (LDR): This is a measure of the difference in returned power between horizontally and vertically polarized radar signals. It is calculated as the ratio of the cross-polar equivalent reflectivity factor ($Z_{vh}$) to the copolar equivalent reflectivity factor ($Z_{hh}$), typically expressed in decibels: LDR = 10 log($Z_{vh}$/$Z_{hh}$). It can provide information about the shape and orientation of hydrometeors. A perfect symmetric particle with respect to the horizontally oriented electric field of the incident radar wave would theoretically result in a $Z_{vh}$ of 0 $mm^6 m^-3$, leading to an LDR approaching negative infinity. However, a finite LDR value implies that there is some degree of asymmetry in the particle relative to the electric field (Doviak et al., 1979; Matrosov et al., 1996). LDR values are influenced by the radar beam's elevation angle; horizontally oriented ice crystals typically show lower LDR values when the radar beam is vertical, compared to columnar or needle-shaped ice crystals with random orientations within the horizontal plane(Oue et al., 2015b). At a vertical incidence, Matrosov (Matrosov et al., 2001) and Reinking (Reinking et al., 2002) observed that columnar ice crystals produced higher LDR values than graupel when using Ka-band radar to observe midlatitude clouds.

## 2.3  GOES-16

GOES-16, launched on November 19, 2016 (Schmit et al., 2017), features the Advanced Baseline Imager (ABI), a 16-band radiometer that significantly enhances the spectral, spatial, and temporal resolution capabilities over its predecessors. The ABI bands encompass visible, near-infrared, and infrared wavelengths, facilitating a broad spectrum of environmental monitoring applications. To observe the storm size and dynamic we will use the CMI_C13 band, which, operates within the 10.1-10.6$\mu$m infrared "clean" longwave window, essential for detailed observations of surface and cloud brightness temperatures ranging from 89.62 K to 341.27 K. This instrument's design enables a mix of rapid-scan, contiguous full-disk scanning, allowing for comprehensive coverage.

## 2.4  Numerical simulation with ARPS model

To analyze the synoptic conditions associated with the case studies, the Advanced Regional Prediction System (ARPS) model, developed at the University of Oklahoma (Xue et al., 2000), was utilized. This model has been previously validated for the Andean region by Flores-Rojas et al. (2019, 2021a), who evaluated ARPS's performance in simulating convective activity. Our model configuration and parameterization closely follow those recommended by Flores-Rojas et al. (2021a), with multi-nested domains at horizontal resolutions of 18 km, 6 km, 2 km, and 0.5 km, and 43 vertical levels to capture the complexity of the Andean topography and land surface processes. The simulations were initiated at 0000 UTC for a 36-hour duration, which includes a 12-hour spin-up period, ensuring model fields are well-adjusted to initial and boundary conditions.

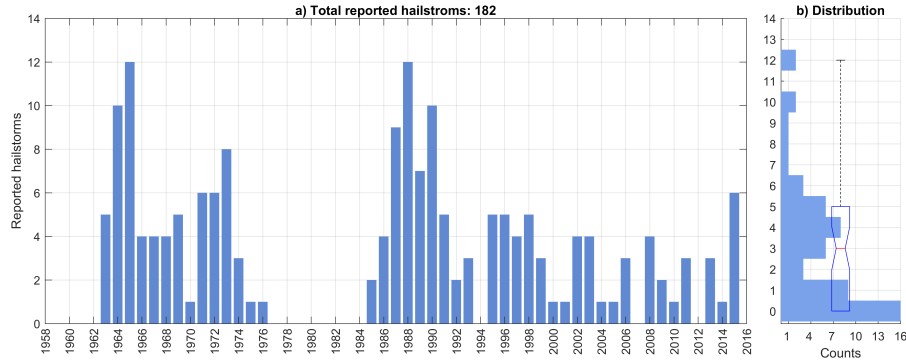

**Figure 2.** a) Historical series of storms reported at the Huancayo Observatory between 1958 and 2016. b) The bars show the histogram of the distribution of reported hailstorms, the same distribution is shown with the box plot with the interquartile ranges.

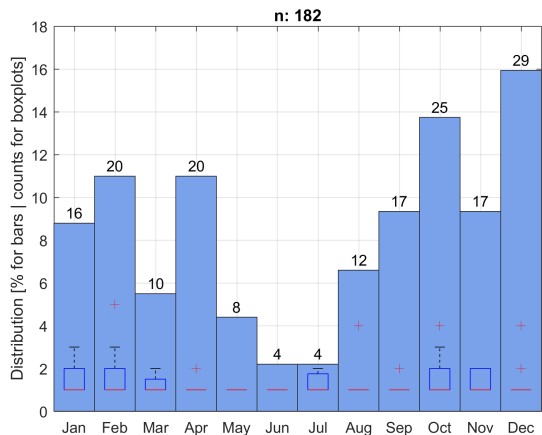

**Figure 3.** Monthly distribution of reported hailstorms, the y-axis shows the relative distribution of all data, while the number above each bar indicates the total number of reports for the month. For each month, the box plot of the annual distribution of hailstorm reports is shown with the respective interquartile ranges. The total number of reported hailstorms is 182.

## 3 Results

### 3.1 Historical hailstorms observations over Huancayo Observatory

The analysis of historical hailstorm reports focused on the number of events, as data on duration were limited. Although the reports date back to 1958, the first recorded hailstorms appeared in 1962. Figure 2a shows the time series of historical hailstorm reports. A significant number of hailstorms occurred between 1963 and 1976, including the highest number of recorded hailstorms in a single year (1965). The number of reports then declined from 1973 to 1974 and reached zero from 1977 to 1984.

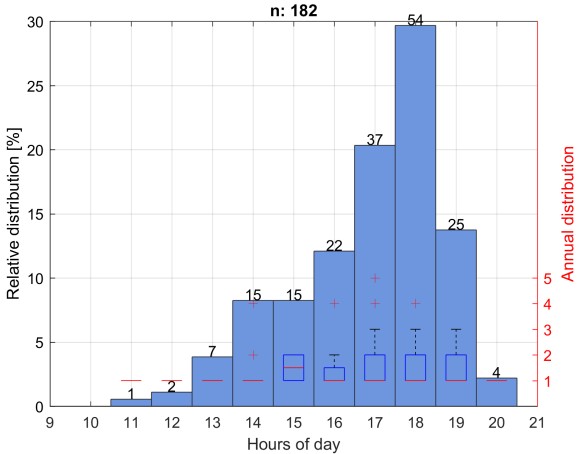

**Figure 4.** Diurnal cycle of reported hailstorms from 1958 to 2016. The y-axis (left) represents the relative distribution of all data, while the number above each bar indicates the total number of cases recorded for that time of day. The y-axis on the right indicates the annual distribution of the records shown in the box plots with the interquartile ranges. The total number of registered events is 182.

A fluctuating cycle of increasing and decreasing events emerged between 1985 and 1993, with the second-highest number of events reported in 1988. No hailstorms were reported in 1994. However, the behavior of hailstorms from 1995 onward significantly differs from previous years, with only 1 to 5 reports per year and minimal variability. The overall average number of annual reports is 3, with 75% of the data (i.e., the third quartile $Q_3$) falling between 0 and 5 reports per year (Figure 2b). Notably, up to 12 events were reported per year in 1965 and 1988. A trend analysis of the data from 1985 reveals a negative trend of -0.5 hailstorms per decade, with a p-value of 0.0737 (Figure 2a).

Figure 3 shows the monthly distribution of reported hailstorms. The inter annual behavior of the reported hailstorms has a similar behavior that the precipitation cycle (Villalobos-Puma et al., 2019; Kumar et al., 2020; Chavez et al., 2020). The rainy season occurs during Austral summer monsoon (December, January, February and March). From September to November is considered as the transition period. June and July are the driest months of the year and also have the lowest historical incidence of hailstorms. Despite the fact that October and December are the months with the highest reported events, it can be seen in the box plot of Figure 2 that January, February and October show the greatest variability in the number of events reported per year, and that at the same time these are months with a high number of historically reported events. In several of the months, only one event per year has been registered, meaning that any number of events greater than one is considered an outlier (Figure 3). Figure 4 shows the diurnal cycle of reported hailstorms. It can be seen that hailstorms are only reported between 11:00 and 20:00 LT. Only 3 events have been registered before 13:00 LT, and almost 20% of events have been registered at 18 LT (54 events). The hours with the greatest annual variability are from 13 to 17 LT, while in other hours only 1 event per year is usually recorded.

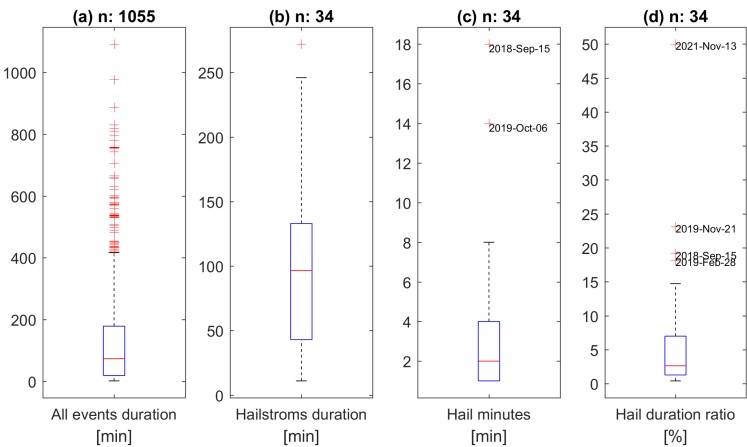

**Figure 5.** Duration distribution of events recorded by Parsivel2. a) Duration of all precipitation events, including liquid and solid. b) Duration of hailstorm events. c) Duration of hail minutes in hailstorms events. d) Relative duration of hail minutes within the event. The box plots show the interquartile ranges.

## 3.2 Disdrometer hailstorms observations

The disdrometer allows us to classify all precipitation events. Here we define an event of precipitation following Tokay et al. (2014) definition. A precipitation event is defined as a period of precipitation separated by 2 hours or longer precipitation-free periods in the rain-rate time series of the disdrometer. Single minute events have been discarded. The rain/no-rain threshold was a set of minimun of 10 drops and a rain rate of 0.1 mm h$^{-1}$. Figure 5 shows the distribution of the duration of the events recorded by the Parsivel2. 1055 precipitation events were found, of which 35 were classified as hailstorm events. Most of the precipitation events are less than 200 min in duration (i.e. between the first and third quartiles, $Q_1 - Q_3$; Figure 5a), however, the whisker in the box plot reaches 400 min and there are several events above that reach up to 1100 min (Figure 5a). Long duration events in the area are related to stratiform events (Villalobos-Puma et al., 2019) that can last all night. Hailstorms last less than 250 min, half of the events last between 40 and 125 min (i.e. between the first and third quartiles, $Q_1 - Q_3$; Figure 5b). Hailstorm events in the area are usually mixed with rain, among the 34 registered hail events, 75% events have 1 to 4 min of hail and two events were found whose duration is 14, and 18 min, respectively. They appear as outliers in the box plot of Figure 5c. In relative terms, the minutes of hail do not exceed 7% of the total event duration in the 75% of the events (i.e. the second quartile is 7%; Figure 5d). There are four events where the hailstorm minutes are above normal, with 18%, 19%, 23%, and 50% of the duration of the event, respectively.

Despite the fact that the observation period of the disdrometer is only from 2017 to 2021, very short to establish a climatology, it is important to see if the behavior of the events is similar to that of the historical reports. Figure 6 shows the monthly distribution of hailstorms detected by Parsivel2. No events were recorded between the months of May and July, which coincides with the dry season and the few events recorded in historical reports. Most months have too few events recorded to be

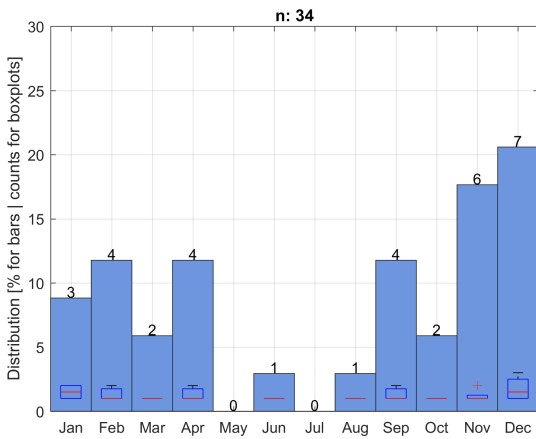

**Figure 6.** Same as Figure 3 but for monthly distribution of hailstorms detected by Parsivel2. The total number of reported hailstorms is 26.

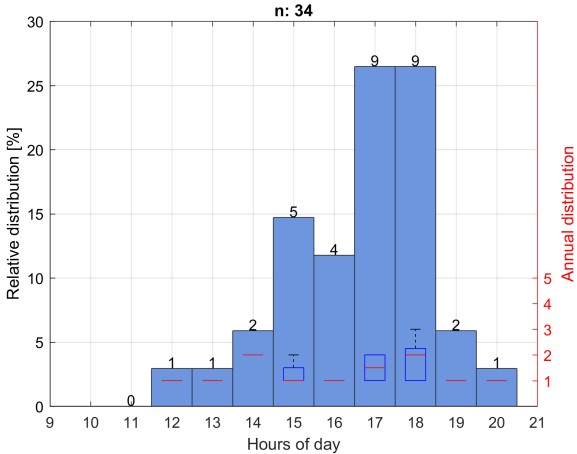

**Figure 7.** Same as Figure 4 but for diurnal cycle of hailstorms detected by Parsivel2. The total number of reported hailstorms is 26.

considered statistically relevant. An increase is observed in the month of September, which would be linked to the transition period. November and December are the months with the highest number of events detected. As in the historical records, the month of April has a higher number of hail events. The box plots in Figure 6 do not give much information since in most

240 months there is only one event recorded. Figure 7 shows the diurnal cycle of hailstorms measured by Parsivel2. The hours in which the events occur are the similar as those reported historically. Hailstorms are only reported between 13:00 and 20:00 LT. Only 1 event have been measured at 13:00 LT, and and all other events after 15:00 LT. 30% of events have been registered at 17:00 LT (8 events). The hours with the greatest annual variability are from 15, 17 and 18 LT, while in other hours only 1 event per year was recorded.

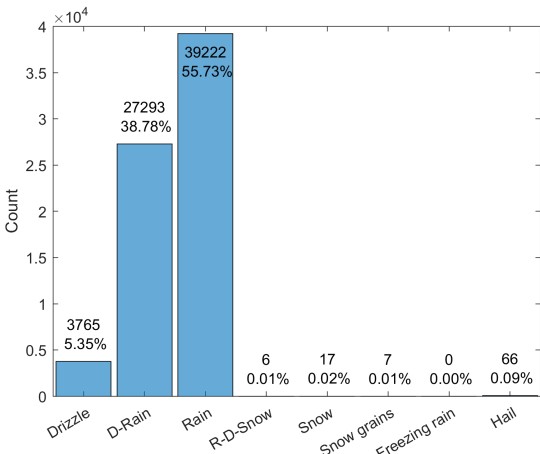

**Figure 8.** Types of precipitation recorded by Parsivel2 according to the Synop code. The y-axis indicates the number of minutes recorded in the entire study period. Above each bar is shown the value of each bar and its relative value.

### 3.2.1 Microphysical characteristics of hail

The types of precipitation recorded by the Parsivel2 have been classified according to the Synop code wawa4860 (WMO, 2019). The categories include drizzle, drizzle with rain, rain, rain drizzle with snow, snow, snow grains, freezing rain, and hail. Theses categories are defined in the internal software of Parsivel2 (OTT, 2016). Figure 8 shows a histogram of the occurrence of different types of precipitation for the one-minute resolution Parsivel2 recording. The most common types of precipitation at the Huancayo Observatory are rain, drizzle with rain, and drizzle, accounting for 55.73%, 38.78%, and 5.35% of events, respectively. This suggests that the majority of precipitation at the observatory is in the form of rain or light drizzle. Only 1.4% of events are from the other categories, with hail representing a very low 0.09%. The minutes of hail in the study area are significantly lower compared to the amount of rain recorded. This is evident in the histogram of Figure 8, where the number of hail events is much smaller than the number of rain events.

Figure 9 shows the relationship between the terminal velocity and the diameter of the particles observed by Parsivel2 for the minutes classified as hail. A total of 66 minutes of hail were identified in the period from 2017 to 2021. The size-velocity matrix and hail size distribution measured with Parsivel2 indicate that hailstones in the area do not usually exceed 14 mm in diameter and they can reach 21 m $^{-1}$. One particle was measured between 14 mm and 16 mm, with less than 1 m s$^{-1}$. Despite the fact that only minutes of hail are shown, it can be seen in Figure 9 that there are also a large number of raindrops present, which follow the theoretical curve of their terminal velocity vs. diameter quite well. It can be difficult to distinguish with the naked eye which data points within the velocity matrix correspond exclusively to hail. No evident hail size-velocity relationship was found. However, we know that raindrops do not usually exceed 8 mm in diameter because they break, nor do they reach very high velocities (Matthews and Mason, 1964; Blanchard and Spencer, 1970; Villermaux and Bossa, 2009). The drops at the Huancayo observatory do not usually exceed 12 m s$^{-1}$ (Valdivia et al., 2020b). From this, we can infer that the block of

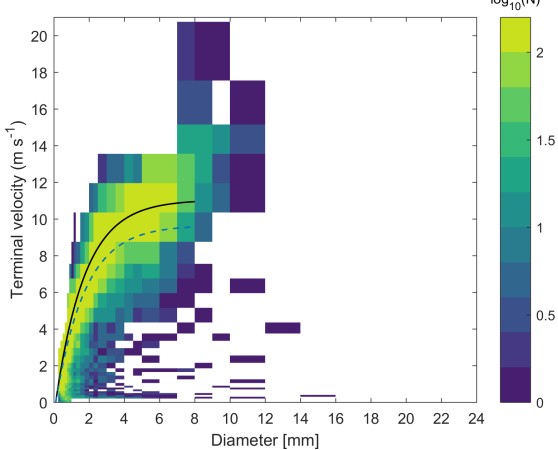

**Figure 9.** Terminal fall velocity as a function of diameter observed by Parsivel2 for the minutes classified as hail (66 min). The theoretical fall velocity for raindrops at 3300 m above sea level is represented by the black line, while the calculation at sea level is depicted by the blue dashed line. It should be noted that this data, while only representing hail minutes, also contains a significant number of raindrops.

particles observed between 8 and 12 mm in diameter and with velocities greater than 10 m s$^{-1}$ may correspond to hail. There are also several particles that are quite outside the typical raindrop distribution and below the theoretical ratio. It is difficult to say for sure if these data points correspond to hail, but it is definitely not normal behavior for raindrops. Valdivia et al. (2020b) conducted an evaluation of the characteristics of the raindrop terminal fall velocity and drop size diameter relationship during a rainy season. Among their findings, it was observed that raindrops are usually distributed quite uniformly close to the theoretical relationship (shown as the black line in Figure 9).

The size distribution of hydrometeors for different types is presented in Figure 10. A noticeable difference is observed between the distribution of liquid rain and hail. For hydrometeors larger than 1 mm, the concentration is higher in hail compared to liquid rain. Hail of up to 15 mm was observed, but the average profile of all the data (represented by the black line in Figure 9) shows particles up to 19 mm. Since 56.7% of the data is classified as rain, its distribution closely resembles the combined distribution of all hydrometeor types. The observed larger particle sizes, which are theoretically improbable for raindrops, might be due to multiple smaller raindrops passing through the Parsivel2's measurement beam simultaneously or high wind velocities, leading to an overestimation of the particle size. This apparently does not generate an artificial classification as hail. The observation of large particles classified as "rain" when it is theoretically improbable for raindrops to be so large may be explained by the work of Friedrich et al. (2013b). Their study suggests that particles with sizes >5 mm and real velocities <1 m s$^{-1}$ can be caused by strong winds and splashes on the head of the instrument, which might result in the overestimation of particle size. The same phenomenon is observed in the size distributions of drizzle and drizzle with rain, where there is an anomalous increase in the largest sizes of each distribution.

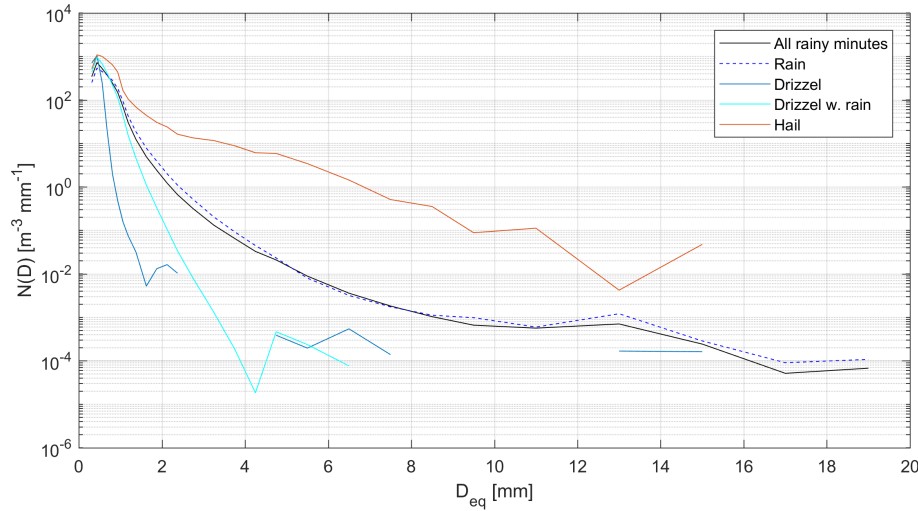

**Figure 10.** Mean particle size distribution for different precipitation types measured by Parsivel2. The black line represents all rainy minutes (124454 min), the dashed blue line represents rain (70548 min), the sky blue line represents drizzle (6782 min), the cyan line represents drizzle with rain (46955 min), and the red line represents hail minutes (66 min).

## 3.3 Vertically pointing radar observations

We conducted a statistical analysis of hail events observed by the MIRA35c radar, analyzing their vertical profiles. To compare the characteristics of hail-only minutes to those of the entire event, we separated the hail profiles using Parsivel2, and the comparison is shown in Figure 11. We observed that the reflectivity profiles of hail minutes had slightly higher values near the surface but rapidly fell as the height increased (Figure 11a). This was primarily due to the attenuation generated by hail and the high intensity of rainfall associated with hail events, as discussed in (Bringi and Chandrasekar, 2001; Peters et al., 2010). It is important to note that we did not apply any attenuation correction algorithm in our analysis. In terms of radial velocity (Figure 11b), which is also the falling velocity as the radar points vertically, we noted a notable concentration of velocities around 10 m s$^{-1}$, which exceeded 12 m s$^{-1}$, surpassing the Doppler Nyquist velocity and resulting in aliasing (Doviak et al., 1979). Unlike the distribution of the entire events, the vertical hail profiles slowly lose velocity with height. At a range of 2 km, it is highly unlikely that aliasing will occur in the falling velocities in the absence of hailstones. The hailstone profiles only show velocities approaching 0 m s$^{-1}$ above the 4 km range. In contrast, for complete events, this occurs before 2 km, where the bright band of the radar or melting layer is approximately located.

In terms of spectral width (Figure 11c), values below the melting layer had a very similar behavior, with the most frequent values around 1 m s$^{-1}$ in both hail-only profiles and the entire event. However, for hail-only profiles, the spectrum was wider towards positive values. Above the melting layer, there was a notable difference in spectral width values, as values in hail minutes did not decrease and tended to increase with height due to turbulence caused by convection in the middle and upper

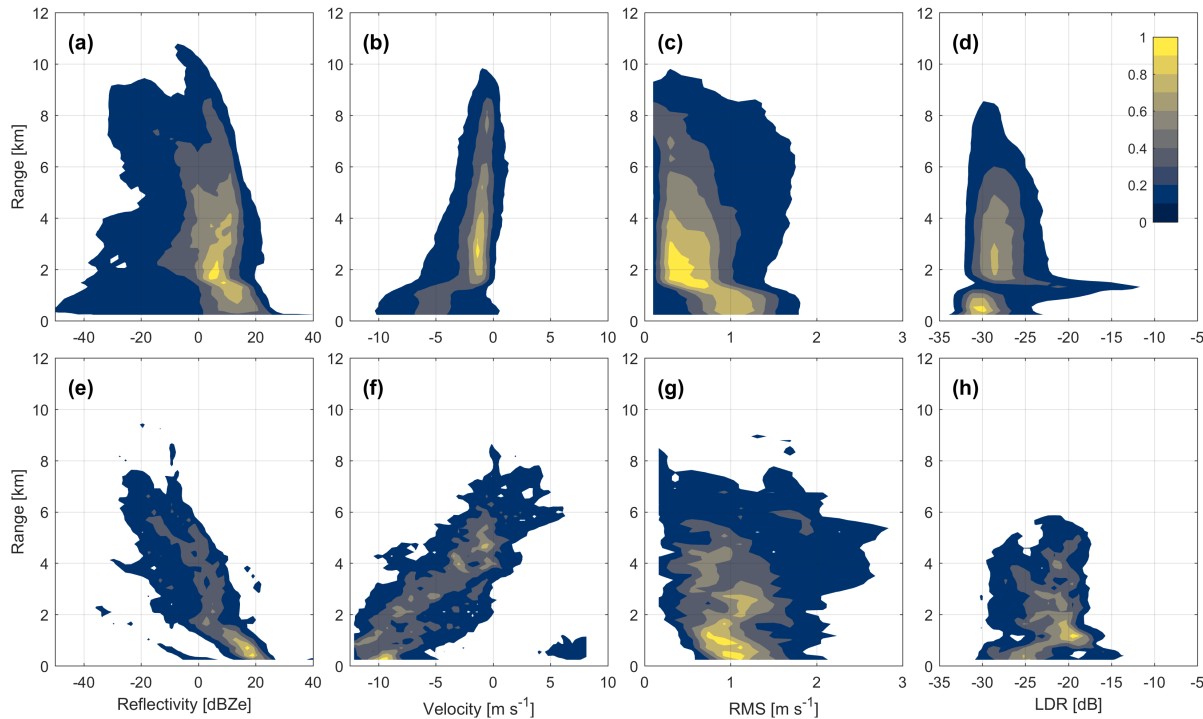

**Figure 11.** Normalized distribution of vertical hail profiles from the MIRA35c radar. Panels (a) to (d) display the overall normalized distribution for all hailstorm events, while panels (e) to (h) specifically show the profiles identified as hail by the Parsivel2 disdrometer. The radar variables presented are: (a, e) equivalent radar reflectivity factor [dBZe], (b, f) radial velocity [m s$^{-1}$], (c, g) spectral width [m s$^{-1}$], and (d, h) linear polarization ratio [dB].

parts of the storm. In the all-event profiles, the mode clusters around 0 m s$^{-1}$, representing the largest amount of data. However, in the hail-only profiles, the mode tends to remain near 1 m s$^{-1}$.

Regarding the linear depolarization ratio (LDR) of the MIRA35c (Figure 11d), there were differences in hail-only profiles. The entire vertical profile of LDR shifted to the right. The LDR for the entire event was around -30 dB near the surface, increasing to values around -28 dB above the melting layer with a slightly wider distribution. For hail-only profiles, the behavior was the same, but approximately 5 dB higher, with differences only in the melting layer where LDR values were lower for hail-only. It is important to note that LDR values are higher in the presence of non-spherical particles, so the melting layer presents high LDR values. Raindrops are expected to have lower LDR values than hail. However, our results show that the LDR of the hail tends to be lower than that of the melting layer. This may be because hailstones are usually accompanied by raindrops, decreasing the total LDR, and the turbulence associated with hailstorm processes typically disrupts the melt layer (Houze, 1993; Williams et al., 1995).

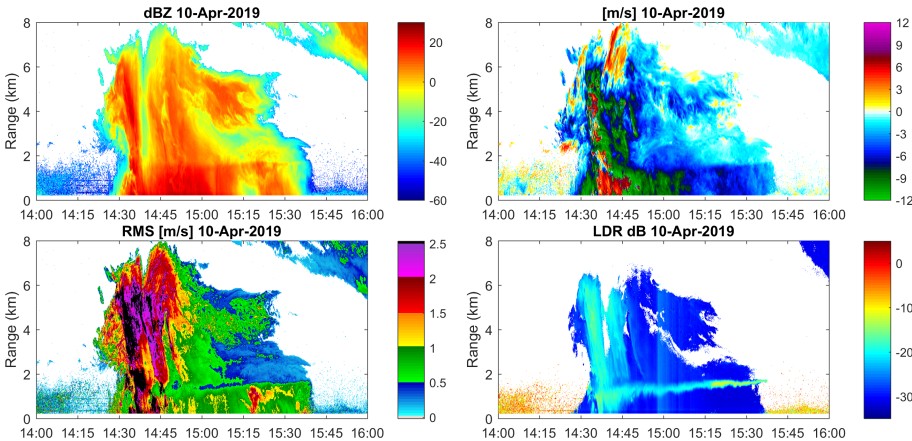

**Figure 12.** Radar variables measured by MIRA35c for Study Case E1 on April 10th, 2019. (a) Equivalent radar reflectivity factor [dBZe], (b) radial velocity [m/s], (c) spectral width [m/s], and (d) linear depolarization ratio LDR [dB] are shown.

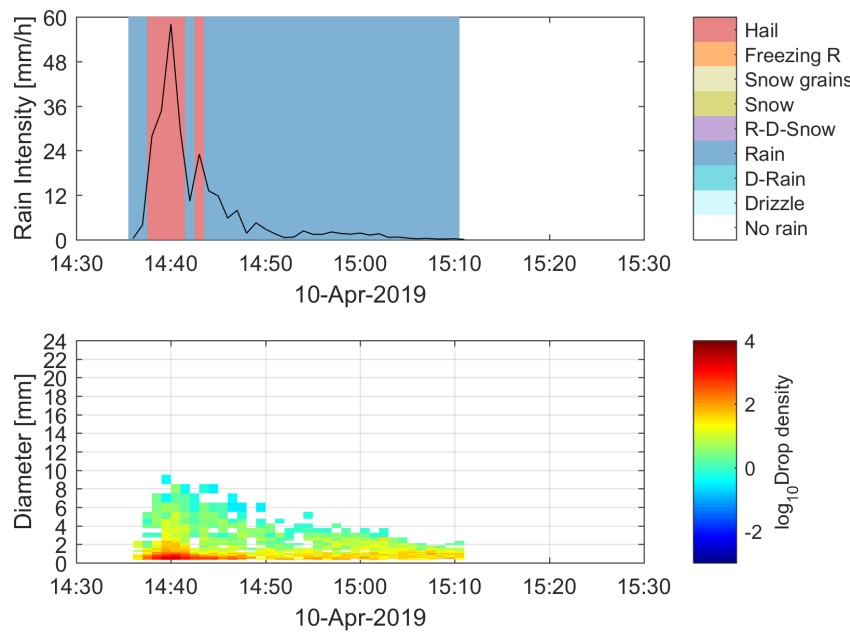

**Figure 13.** Parsivel2 observations for Study Case E1 on April 10th, 2019, local time, (a) shows the rain intensity [mm/h] is shown in black lines along with the Synop 4680 hydrometeor classification in shading color, and (b) shows the hydrometeor size distribution [m$^{-1}$ mm$^{-1}$].

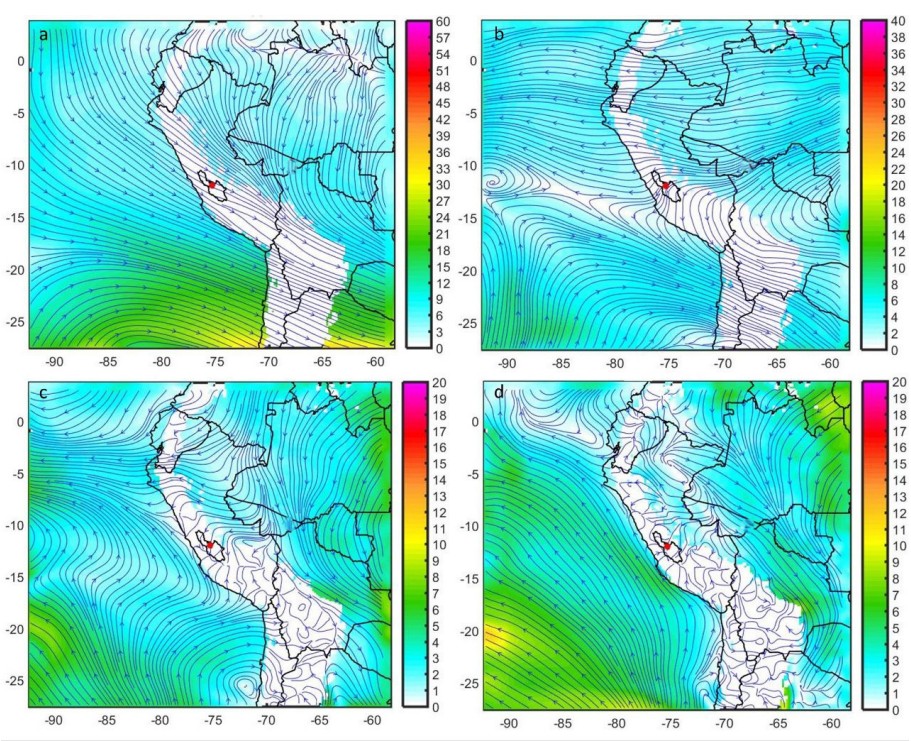

**Figure 14.** Composites of horizontal wind streamlines and wind intensity (m s$^{-1}$) on 10 April 2019 at 15 LT (20UTC) for (a) 250 hPa, (b) 500 hPa, (c) 700 hPa and (d) 850 hPa. All fields were obtained with the ARPS model for the resolution of 18 km. The location of the Huancayo Observatory is indicated by the red point. Longitudes, latitudes, and contours of the Mantaro basin are indicated.

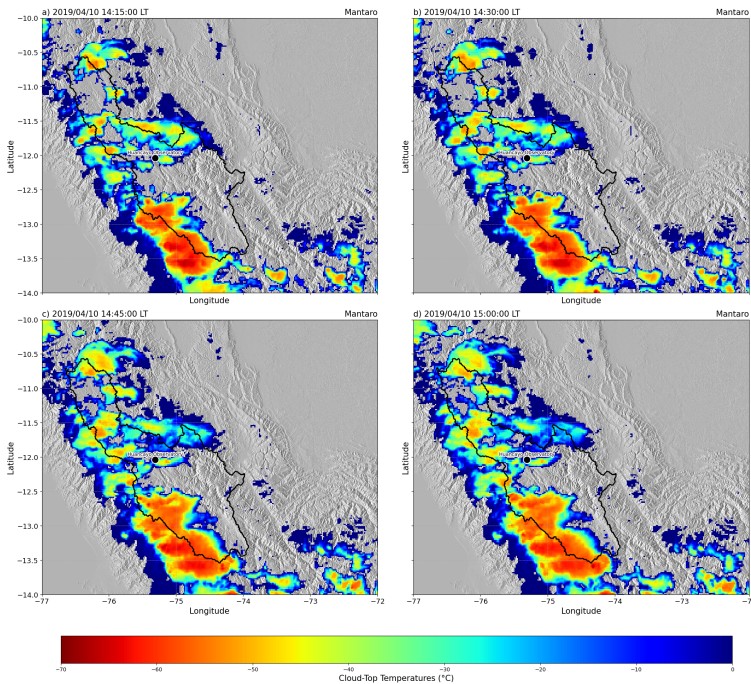

**Figure 15.** GOES-16 brightness temperature (10.1-10.6 $\mu$ m) for the study case of April 10th 2019 (E1) at a) 14:15 LT, b) 14:30 LT, c) 14:45 LT and d) 15:00 LT.

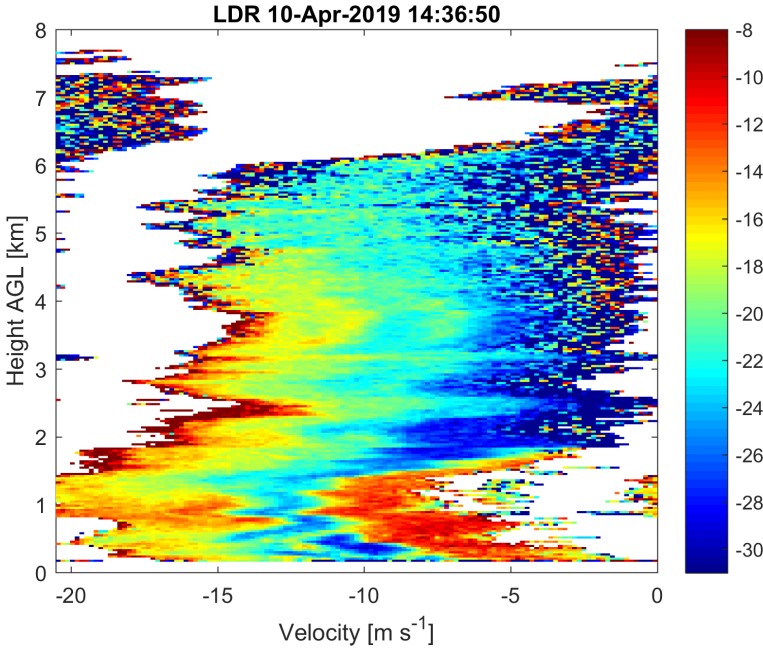

**Figure 16.** Spectral LDR from MIRA35c Doppler spectra at 14:36:50 April 10th 2019 (E1).

## 3.4 Study cases

In order to assess the distinctive characteristics of hailstorm, we selected two study cases characterized by distinct radar signatures. The first event was chosen because it exhibited strong radar signals, suggesting the possibility of it being a hail event. On the other hand, the second event did not exhibit clear radar indications of hailfall.

### 3.4.1 Case April 10th, 2019 (E1)

The case study of April 10th, 2019 (E1) has notable radar signatures. On this day, the MIRA35c radar (Figure 12) recorded high values of reflectivity, velocities and spectral width accompanied by high attenuation. The Parsivel2 disdrometer indicated the presence of hail (Figure 13) for approximately 6 minutes around 14:40 LT.

The synoptic setup for E1 was characterized by a complex interplay of atmospheric circulations at different pressure levels. At 250 hPa high-level (Figure 14a), a robust westerly flow prevailed, which contrasted with the conditions observed at 500 hPa mid-level (Figure 14a), where an easterly circulation dominated within the valley. Simultaneously, lower levels at 700 hPa and 850 hPa exhibited strong southwesterly flows, facilitating the introduction of moist air from the Pacific into the Andean region, as depicted in Figure 14c,d. These dynamic conditions, combined with the influence of the South America Low-Level Jet at both 700 hPa and 850 hPa levels, facilitate the moisture transport from the equator to the higher latitudes of the Mantaro basin.

GOES-16 imagery provided a view of the convective processes associated with E1. Initially, at 14:15 LT (Figure 15a) before precipitation is observed on the surface there are no features of a convective system over the Huancayo Observatory (Figure 15b). At 2:30 p.m., where the precipitation began, it can be seen that the convective core is small just above the radar location and that it dissipates quickly in an easterly direction. This suggests that the convection occurred just above the observatory, on a small scale and of short duration.

The Parsivel2 data showed that the hailstones fell for approximately 6 minutes around 14:40 LT, a few minutes after the start of the rain. The total duration of the event was approximately 35 minutes. The first 15 minutes of the event presented convective characteristics with high rain intensity, which then evolved into a stratiform precipitation. Within these first 15 minutes of E1, Parsivel2 classified the precipitation peaks as hail (Figure 13a), with particle diameters greater than 6 mm (Figure 13b). High values of reflectivity and attenuation were observed in Figure 12a, especially during the minutes of hail. The fall velcoity during the minutes of hail was quite high throughout the vertical profile (Figure 12b), with downward velocities of 12 m s$^{-1}$ from 6 km above the radar to the surface, surpassing the Doppler Nyquist velocity in several sections. Based on the distribution presented in Figure 11c, we can infer that spectral width values above 2.5 m s$^{-1}$ could be presented as outliers. The LDR values during the minutes of hail presented values around -20 dB (Figure 12d), which is coherent with the distributions presented in Figure 11d in the presence of hail. It is interesting to note that the high LDR values are shown from a high altitude, covering practically the entire vertical profile of precipitation. Considering that the fall velocities are also high at high altitude, it can be inferred that hailstones are present even above 6 km from the radar.

Figure 16 shows the spectral LDR at 14:36 LT of the E1 event. The depicted velocity spectrum exhibits a pronounced zigzag pattern, suggestive of turbulent activity within the atmospheric column. Notably, at an altitude of 1 km AGL, velocities surpass

the 20 m s$^{-1}$ mark, likely enhanced by the influence of turbulence. At altitudes closer to the surface, velocities reach up to

17 m s$^{-1}$. It is important to highlight that the Nyquist downward velocity parameter set for the MIRA35c radar in this study
is -12 m s$^{-1}$. The Doppler spectrum of this particular event represents a challenge in terms of radar configuration, due to the
wide range of velocities observed. This includes upward velocities (positive) peaking at 5 m s$^{-1}$ observed between altitudes
of 6 km to 7.4 km AGL, and descending velocities (negative) of up to 22 m s$^{-1}$ detected between 0.8 km and 1.5 km AGL.
The most significant LDR values, peaking at -8 dB, correspond with the highest downward velocities, which is consistent with

the observation of hail from 5km AGL to the surface. Moreover, at altitudes below 2 km AGL, substantial LDR values of
approximately -11 dB are observed in conjunction with the lowest fall velocities. These low-velocity, high-LDR signatures are
inconsistent with typical hail velocities and may be associated with the melting layer and downdrafts.

### 3.4.2   Case March 7th, 2019 (E2)

The second case study, E2, occurred on March 7th, 2019 (Figure 17). Unlike E1, there were no evident values in the MIRA35c

radar variables suggesting that we were in a hail event. However, the Parsivel2 data shows that there were three minutes of hail
just before 14:30 LT (Figure 18a).

The synoptic conditions on E2 (Figure 19), revealed distinct atmospheric circulations at various pressure levels. At the 250
hPa high-level (Figure 14a), a robust easterly circulation indicative of the Bolivian high formation was observed over the
Peruvian central Andes. Moving to the mid-level at 500 hPa (Figure 14b), a strong eastward flow traversed the Mantaro basin,

synchronized with the circulation pattern at 250 hPa and influenced by a prominent anticyclonic circulation centered over the
Pacific Ocean at 20°S, 71°W. Meanwhile, at the 700 hPa level (Figure 14c), a vigorous south-westerly circulation enveloped
the location of the Huancayo Observatory. Further down at the 850 hPa level (Figure 14d), notable circulations originating from
the Pacific Ocean infiltrated towards the Andes cordillera, particularly evident near the surface at the latitude of the Huancayo
Observatory. Additionally, the presence of the South America Low-Level Jet (SALLJ) on the eastern flank of the Andes at both

700 hPa and 850 hPa levels facilitated the transport of moisture-laden air from the equator towards the higher latitudes of the
Mantaro basin.

The sequence of GOES-16 satellite observations for event E2 provides a detailed chronology of the convective activity in the
study region. At 14:00 local time, the initial stages of E2 were observed northeast of the Huancayo Observatory (Figure 20a),
with the system developing and moving southwestward, eventually passing over the MIRA35c radar by 14:15 LT. The intensi-

fication of the system continued, and by 14:30 LT (Figure 20b), the convective core was still developing, coinciding with the
detection of hailstones by the Parsivel2 instrument. Approaching its mature phase around 15:00 LT (Figure 20c), the convective
core's maximum intensity was reached. However, the MIRA35c radar was primarily detecting only the periphery of the storm.
By 15:30 LT, the system began to dissipate, spreading over a broader area (Figure 20d).

In the Parsivel2 drop size distribution (Figure 18b), there are no notably larger drops, only a few minutes exceeded 6 mm

in diameter, but not exceeding 8 mm. E2 was observed as a convective precipitation of approximately an hour in duration.
The convective part occurred in the first 30 minutes of the event, and then it transformed into a more stratiform rainfall. Near
14:30 LT, when hail was recorded, attenuation in radar reflectivity was observed (Figure 17a). The radial velocity was quite

high during the convective minutes, even exceeding the Doppler Nyquist velocity, reaching up to 12 m s$^{-1}$ (Figure 17b). Although hail was recorded in the first few minutes of the event, the spectral width appears to be quite homogeneous, with small groups disintegrated between 2 and 4 km, with values between 1.5 m s$^{-1}$ and 2 m s$^{-1}$ observed at the beginning of the event. Such values are typical in the presence of raindrops. However, below 2 km, the spectral width values are much more homogeneous (Figure 17c), between 0.5 m s$^{-1}$ and 1 m s$^{-1}$, which correspond to more typical values of light rain or cloud values (Figure 11c). These observations of uniform spectral width and high fall velocities suggest an absence of turbulence and hail and raindrop sizes quite similar, except for at small portion around 0.7 km AGL at 14:27 LT. The LDR values in this part of the event show values close to 20 dB (Figure 17d), which is consistent with the presence of hail, based on the observations of the variable distribution shown in Figure 11d. When the convective system begins to become more stratiform, that is when upward velocities and high turbulence are observed (Figure 17b,c). However, the LDR values show that hail formation starts much earlier while the core is moving southwestward.

Figure 21 shows the spectral LDR at 14:27 LT of the E2 event. A pronounced zigzag pattern can be observed over 1.5 km, and only one peak velocity at 0.7 km. In this Doppler spectra a side-lobe effects can be observed near the surface at the right-side of the main echo. Unlike the E1, in this event the velocities are lower. The peak velocities at 0.7 km AGL, does not surpass the 17 m s$^{-1}$. Closer to the surface, velocities reach up 12.5 m s$^{-1}$. The spectral LDR, shows higher values than E1, even over -8dB, between 0.5 km and 2.5 km AGL, but these high LDR disappears near the surface, likely because the hailstones are being advected outside of the radar sight. Close the surface the highest LDR values are -22 dB, still consistent with hail range in Figure 11, with velocities of 10 m s$^{-1}$.

## 4  Discussion

The present study aimed to analyze the characteristics of hailstorms in a specific region using both ground-based and radar measurements. The analysis showed a decrease in the frequency of hailstorms over time, at a rate of -0.5 per decade, and this finding is based on observations from a single location. While this trend may not be statistically significant, it is close to the threshold and thus warrants further investigation. It is also unclear whether this trend is due to climate change or changes in reporting methods. Changes in the way hail is reported could lead to errors in these results. On the other hand, negative trends in the frequency of hailstorms between Argentina and Brazil (Beal et al., 2020) are associated with an increase in the melting layer, which in turn is linked to global warming (Xie et al., 2008).Further studies are necessary to determine whether there have been any changes that could lead to a negative trend in hailstorm frequency.

We found significantly higher number of events per year in the Parsivel2 observations (8.5 events/year) that the historical reports (3 events/year). This discrepancy could be due to the fact that in the historical data there are many events that have not been recorded, either because the observer did not realize it, or was not present at the time of the occurrence. Most of the events found in this study last just a few minutes, and as seen in the case studies, the events could be quite focused.

Microphysically, the study found few differences that could be used to identify hail particles. Most particles in the main distribution of drops in the velocity versus diameter matrix were easily counted and were very close to or equal to one. The

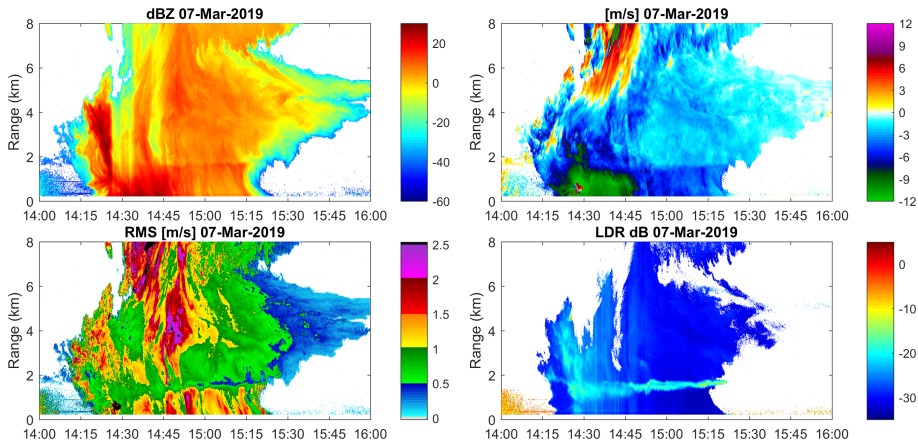

**Figure 17.** Same as Figure 11, but for E2 of March 07th, 2019.

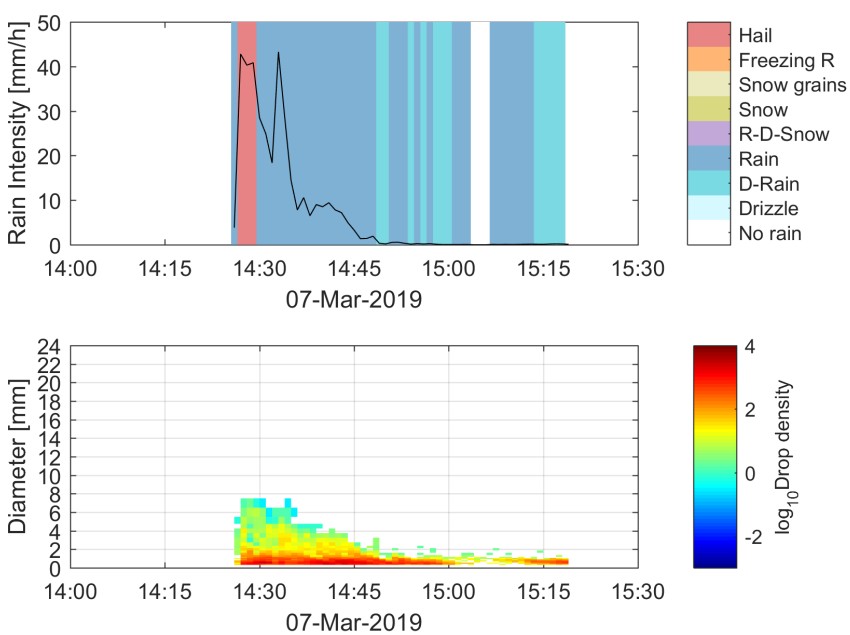

**Figure 18.** Same as Figure 12, but for E2 of March 07th, 2019.

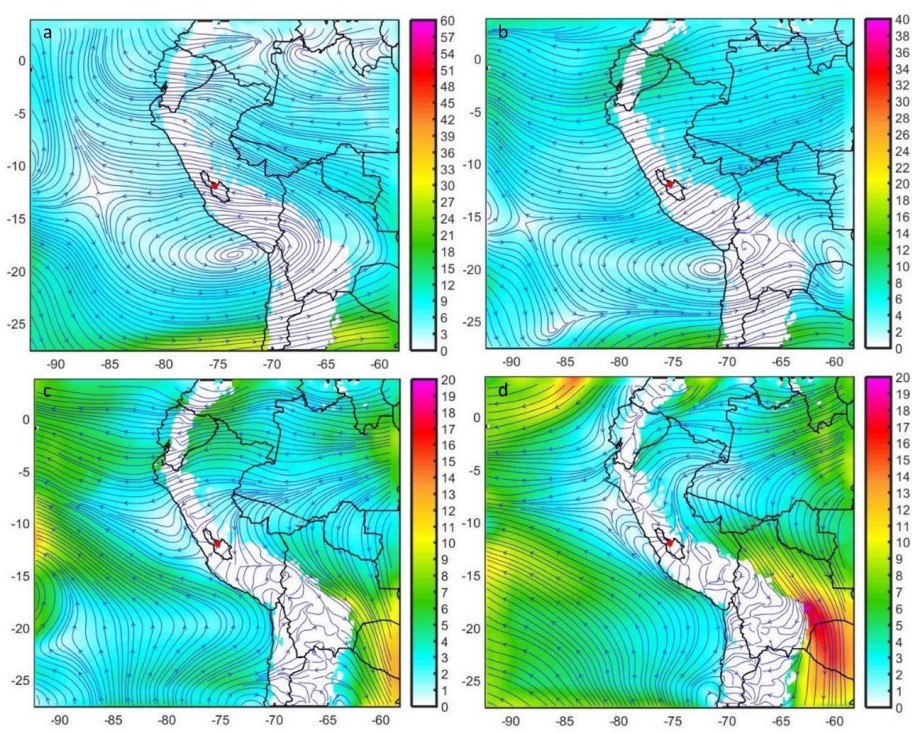

**Figure 19.** Same as Figure 14 but for 07 March 2019 (E2).

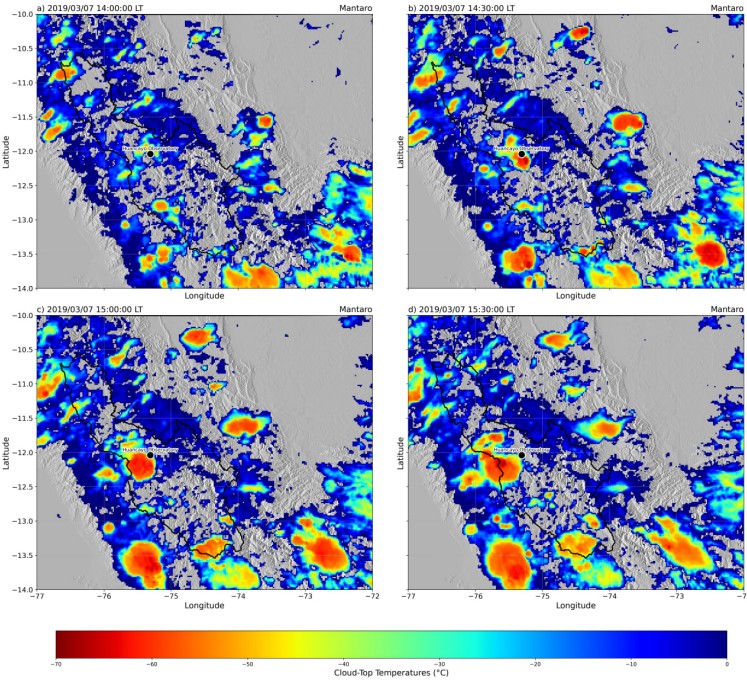

**Figure 20.** Same as Figure 15 but the study case of March 7th at a) 14:00 LT, b) 14:30 LT, c) 15:00 LT, and d) 15:30 LT.

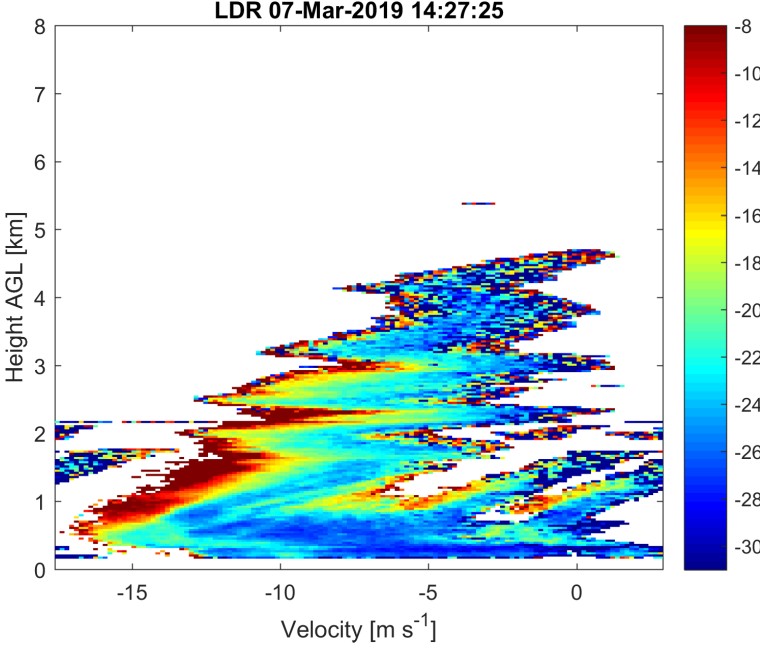

**Figure 21.** Spectral LDR from MIRA35c Doppler spectra at 14:36:50 March 7th 2019 (E2).

PSD showed in Figure 10 is evidence that the Parsivel2 classification of hydrometeors is not perfect. One of the most concerning details is that the largest particle sizes recorded by the Parsivel2 correspond to "rain" precipitation, as seen in Figure 9. The DSDs that correspond to liquid rain have droplet diameters that physically should not exist. This error could be evidence of multiple drops traversing the beam simultaneously or hail being marked as rain. In the case of the estimation of microphysical

properties in rain. Friedrich Friedrich et al. (2013b) suggests that particles with sizes >5 mm and real velocities <1 m s$^{-1}$ can be caused by strong winds and splashes on the head of the instrument, which might result in the overestimation of particle size. In our study, it is possible that similar conditions have led to the presence of large particles classified as rain in the Parsivel2 dataset. To account for this, future research could consider implementing additional measures to mitigate the impact of wind-induced splashes or improve the classification algorithm's ability to distinguish between true raindrops and other sources of

large particles. Studies focused on the estimation of liquid precipitation suggest applying certain filters to correct the Parsivel2 data, however, in this work no filter was applied due to interest in studying the complete velocity-diameter matrix. It is also important to note that the definition of hail used by meteorological observers may differ from the technical definition of hail, which is defined as ice pellets greater than 5 mm in diameter (WMO, 2017). The present study did not discriminate based on size, instead focusing on how the Parsivel2 algorithm classified the hydrometeors. This Parsivel2's classification method is

not well-documented and may lead to errors. The classification of hydrometeors, particularly differentiating hail from graupel, presents a significant challenge in this study due to their similar physical characteristics, which often makes clear identification elusive. Parsivel2, the instrument employed for the hydrometeor classification, does not differentiate between these two forms of solid precipitation. Consequently, the microphysical properties of hail as inferred from Parsivel2 data may be confounded with those of graupel.

The detection of hail using radar variables has proven to be challenging. The reflectivity parameter experiences a rapid decay in the presence of intense precipitation and hailstorms. Additionally, the Nyquist velocity places a constraint on the radial velocity parameter, which poses a significant inconvenience when trying to detect hail. The Nyquist effect also impacts the spectral width parameter, generating artificially inflated values. Within the scope of this study, we have implemented a fundamental dealiasing procedure which involves the transposition of a segment of the spectrum from positive to negative

fall velocity values. This rudimentary method has generally proven sufficient for the observation of rain and snow (eg. Valdivia et al. 2020b; Kneifel et al. 2011). However, more sophisticated techniques are documented in the literature, such as the oversampling of spectra to effectively elevate the Nyquist velocity, a method detailed by Maahn and Kollias (2012). These advanced approaches are designed to address the complexities arising in scenarios where higher velocity ranges are encountered might be necessary for hail observation. The linear depolarization ratio (LDR) stands out as the most reliable indicator of

hail. However, observations in this work suggest that hail is typically accompanied by raindrops, and at times, precipitation of rain may dominate the LDR parameter. To study hailstones accurately, the most effective approach is to analyze the Doppler spectrum directly. The spectral LDR analyzed in our study cases stand as the most reliable technique to identify hailstones and other hydrometeor types. Multiple peak processing to Doppler velocity spectra (e.g. Williams et al. 2018) during storms could help to automatize hail detection. However, the Doppler spectrum is typically computationally intensive and requires detailed

event-by-event analysis. A computationally less expensive technique also at Ka band has been proposed by Sokol et al. (2018),

although hydrometeor classification has not been directly tested in their study due to the absence of a reference instrument, simpler techniques can be derived calculating thresholds from the spectral LDR.

Both case studies present characteristics of very small convective activity. While the case E1 a barely noticeable cloud size in the GOEs-16, the event was develop just over the radar location, so the radar signal are strong. In case E2, a bigger connective core was develop but it was advected and the radar was primarily detecting the periphery if the storm, is likely that the hail stones were also being advected outside the radar sight leading to a very short hail episode over the Huancayo Observatory. In both cases the spectral LDR shows good information about the hailstone location and the effect of turbulence in the radar signals. This information combined with scattering calculation can provide valuable information to study microphysical processes (Mróz et al., 2021; Oue et al., 2015b, a).

## 5 Summary and conclusions

This study aimed to investigate the climatological characteristics of hail events in the Central Andes of Peru. The research utilized historical data from hailstorm records dating back to 1958, as well as four years of observations using Parsivel2 disdrometer and a cloud profiler radar. The focus was on understanding the microphysics of hail and identifying which radar microphysics are associated with hail. We found evidence of a trend in decrease hail frequency (-0.5 events/decade, with the p-value of 0.07 suggest that the observed trend may not be statistically significant, but it is close to the threshold and warrants further investigation). It is not clear if the trends are the product of environmental aspects such as the increase in the melting layer (Beal et al., 2020; Xie et al., 2008), or if they are human errors induced by changes in the way hail is reported. Such findings need to be evaluated in more detail, aspects regards to environmental changes are important to fully understand trends in hail frequency.

We use the Parsivel2 to classify the types of hydrometeors in the period from 2017 to 2021. Parsivel2 classifies hydrometeors using the weather code SYNOP wawa4860. The internal Parsivel2 algorithm of classification is not documented. The results for both, the historical report and Parsivel2, showed that hail events in the region are most common during the austral summer months, with a peak in December. Hailstorms tend to occur during the afternoon and evening hours, with a duration of hail-only minutes is most of cases less than 4 minutes. All reported hail events occurred between 10 am and 9 pm LT, and the average number of reported hailstorms is 3 per year. However, Parsivel2 only flagged as hail the 0.12% of the whole dataset. The size-velocity matrix and hail size distribution measured with Parsivel2 indicate that hailstones in the area do not usually exceed 14 mm in diameter and the can reach 20 m s$^{-1}$. Hail usually falls with raindrops, which is predominant in the diameter-velocity matrix. The conditions that determine the size of the hailstones are still unknown, although we found evidence of a decrease in the frequency of hailstorms, more research is necessary to know how the size of hailstones will change in the future. Note that increasing their size would imply that they would be more destructive than they already are. Understanding these hailstone characteristics set the stage for a deeper radar-based analysis of hailstorms.

A vertically pointing radar is used to study hailstorm characteristic and four variables are used: reflectivity, velocity, spectral width, and LDR (see Section 2.2). The study shows that hail events have distinct vertical profiles in terms of reflectivity, radial

velocity, spectral width, and LDR. The hail profiles have higher reflectivity values near the surface, high falling velocities, and wide spectral width values above the melting layer, indicating turbulence caused by convection in the middle and upper parts of the storm. The linear depolarization ratio of hail-only profiles is also higher than that of the entire events. Our observations indicate that hail is often accompanied by raindrops, which complicates its detection. The LDR is a good indicator of the presence of hail, typical values of LDR profiles for hail found in this study are greater than -27 dB. Despite the relative unreliability of other radar variables in hail detection, our findings indicate discernible differences in the fall velocities typically associated with rain compared to hail. Hail, as detected by radar, generally exhibits average fall speeds of approximately 10 m s$^{-1}$ near the surface; however, these measurements are notably prone to the effects of turbulence. This turbulence significantly impacts both the velocity readings and the spectral width of the hail profiles. Notably, spectral width values exceeding 2 m s$^{-1}$ are likely indicative of an artifact resulting from aliasing. Such a finding serves as a valuable metric, suggesting instances when data might require more stringent quality control to ensure accuracy.

Two case studies were analyzed in detail. Both cases of hailstorm present very different values in the radar variables, and show how diverse hail can be in terms of radar measurements. Both E1 and E2 featured minor convective activities; E1 had a small cloud size visible in GOES-16 imagery, yet it developed directly over the radar, producing strong signals. In contrast, E2 developed a larger convective core that was quickly advected, leading to the radar primarily detecting the storm's periphery. It is probable that the hailstones were also advected away from the radar's view, resulting in a brief hail episode at the Huancayo Observatory. In the presence of intense precipitation and hailstorms, the reflectivity parameter undergoes rapid attenuation, limiting its usefulness. Furthermore, the Nyquist velocity constraint complicates the use of the radial velocity parameter for hail detection, leading to potential underestimation of hailstone velocities. Our first case study showed that the fall velocities in hail can be as high as 20 m/s influenced by turbulence.

In our study, a fundamental dealiasing method was applied, shifting part of the spectrum from the positive to the negative fall velocity domain. This technique, while typically adequate for rain and snow observations may not suffice for accurate hail detection. Advanced methods such as spectral oversampling, which can effectively increase the Nyquist velocity (Maahn and Kollias, 2012), may be necessary to address the challenges presented by higher velocity ranges associated with hail.

The most effective strategy for hailstone analysis is the direct examination of the Doppler spectrum. In both E1 and E2, the spectral LDR provided robust data for identifying hailstones and assessing the turbulence's influence on the radar signals. The implementation of multiple peak processing to Doppler velocity spectra (Williams et al., 2018) could enhance the automation of hail detection. However, such analysis is computationally demanding and necessitates an event-by-event examination. An alternative, proposed by Sokol et al. (2018), offers a less computationally intensive approach at the Ka-band, yet direct testing for hydrometeor classification in their study was not feasible due to the lack of a reference instrument. Nonetheless, simpler techniques might be derived by establishing thresholds from the spectral LDR. When combined with scattering calculations, as discussed in Mróz et al. (2021) and Oue et al. (2015b, a), this information can be instrumental in furthering the understanding of microphysical processes. It is through such detailed and nuanced analyses that we can improve our approaches to studying hailstorms and their associated processes.

Future research to enhance our understanding and detection of hail events is needed, particularly in the complex terrain of the Central Andes of Peru. A thorough review and refinement of the Parsivel2 algorithm are essential. Future studies should aim to improve the accuracy of hydrometeor identification. Determining the specific conditions that lead to the misclassification or overestimation of hydrometeor sizes by Parsivel2 will be necessary. Developing corrective algorithms or filters to improve the accuracy of size and velocity measurements is an important step forward. Addressing the challenges posed by the radar's Nyquist velocity will be crucial. Research should focus on the development and implementation of advanced dealiasing techniques, such as spectral oversampling, to better capture the full range of velocities associated with hail. Given the promise shown by LDR in hail detection, further research should concentrate on optimizing the use of LDR, establishing clearer thresholds for hail detection, and exploring the potential of LDR in differentiating hail from raindrops. Expanding the use of scattering calculations to understand the microphysical properties of hail more deeply would provide valuable insights into the formation and evolution of hailstones. A systematic approach to analyzing a larger set of hailstorm case studies over various seasons and years could reveal patterns and anomalies in hailstorm behavior.

*Data availability.* The data used in this work can be acquired upon request in case of historical report and radar, or download from the repository in the case of Parsivel2 (Valdivia et al., 2020a): https://www.igp.gob.pe/programas-de-investigacion/ciencias-de-la-atmosfera-e-hidrosfera/facilidad/microfisica/acceso-a-datos

*Author contributions.* J.M.V., D.G., E.V.P., J.L.F.R., S.C., and Y.S.V. conceptualized the study. D.G., E.V.P., S.C., and Y.S.V. provided the resources. J.M.V. and D.G. conducted the formal analysis. J.M.V. developed the software and visualization. S.C. and Y.S.V. secured the funding and administered the project. J.L.F.R. and Y.S.V. supervised the project. J.L.F.R., S.C., and Y.S.V. carried out the investigation. D.G., E.V.P., S.C., and Y.S.V. validated the results. J.M.V. wrote the original draft of the manuscript. D.G., E.V.P., J.L.F.R., S.C., and Y.S.V. reviewed and edited the manuscript.

*Competing interests.* The authors declare that they do not have conflict of interest.

*Acknowledgements.* Special thanks to Mr. Gaudencio, who was for many years the meteorological observer at the Huancayo Observatory, his work made this research possible. This work was done under the project "TAMYA - *Impactos de la precipitación, registrados con un radar meteorológico, en los cuerpos glaciares Andinos: nevado Huaytapallana*, CONCYTEC - 082-2021-FONDECYT". We also want to thank to: Gobierno Regional de Junín and Oficina desconcentrada de la región centro de Lima - INAIGEM. We acknowledge the use of OpenAI's GPT, an artificial intelligence language model, during the drafting of this manuscript. Its assistance was primarily utilized for helping with language structure, and enhancing the overall quality and clarity of the text. However, the final interpretations and conclusions presented in this manuscript are solely those of the authors.

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
