# Peer review of "Hailstorm Events in the Central Andes of Peru: Insights from Historical Data and Radar Microphysics"

_EGUsphere, 2023_

## Referee Comment (RC1)

**Review: Hailstorm Events in the Central Andes of Peru: Insights from Historical Data and Radar Microphysics**

**General comments:**

1. The limitations and underlying assumptions for the data and instruments could be discussed more explicitly. It is unconventional to use a cloud radar for a convective study, partially because of attenuation. While this is part of the novelty of this study, this also warrants a discussion. Moreover, there is little to no discussion about the hail detection of Parsivel2 or the reliability of the observer hail reports.
2. There is a lack of connection between the discussion of the long-term hail reports and the microphysical assessment of 2 cases. What is the ulterior motive of diving into these 2 cases in particular? Should the detailed evaluation build the foundation for a later hail-classification algorithm the vertical profiles? This could be a clear motivation in the direction of method-development. The two cases alone are also not enough to make a general statement about the overall diversity of hail profiles.
3. Both case studies would benefit from a (brief) synoptic discussion of their cases. Is there satellite data available to e.g. estimate convective mode or storm structure? Where did the storms originate and how was their track oriented? How was the synoptic weather situation, are they related to fronts? The overall storm characteristics and atmospheric conditions impact hail development and can help understand the differing evolution of the events.
4. There is a lack of contextualization of the results. A discussion of which findings are in line with previous research and which aspects contradict other studies would be very helpful. Partially the suggestions in the outlook have been implemented in other research (Sokol et al., 2018 -> convection and hail in a Ka-band-radar; Wang et al., 2023 -> hail classification on a satellite-borne radar), this at least warrants a mention. Studies focusing on winter precipitation also go into more detail on hydrometeor classification and microphysical understanding with cloud radars and could be used as a comparison. These also often address issues mentioned here (e.g. dealiasing a spectrum).

**Specific comments:**

1. Line 44: Space missing before bracket
2. Line 51: Climatological trends depend a great deal on the length of observations and tend to only properly emerge after >40 years. Hence conflicting results are often an artefact of the length of the timeseries and it would be helpful to include this duration here.
   Consider including the assessment of Taszarek et al. (2021) on changes in severe convective environments.
3. Line 53: Twice Beal
4. While this is not emphasized in the introduction, convective studies in complex terrain and at high altitudes are rather rare and deserve particular emphasis. How does this compare to other complex terrain studies and hail trends in other mountain regions?
5. Section 2: To introduce the region it would be great to have a map of the area. Are the observed hail reports gathered for a larger area than just the observatory? What is considered the representative area for these reports and how does it compare to the area of the Parsivel2?
6. Line 128: What is the representative measurement area of a Parsivel2?

7. Line 142: Consider including the specification Ka-band
8. Line 162: Hail can be very spherical, especially while it is smaller. However, LDR is also influenced by the mix of phases and density in the hailstones.
9. Section 3.1: Is there any assessment of how robust the hail reports are? What were the observation criteria? How does the overall annual hail frequency compare to global hail frequency assessments (Prein, Taszarek, Raupach, …)?
10. Fig. 1 and following: please define the lines of the boxplot (mean + standard deviation / median + quartiles / …)
11. Line 185 (and throughout the manuscript): Please consider using a more precise time format that includes the time zone. Moreover, "hours" usually refers to a duration and not a time of day.
12. Line 189: twice Tokay
13. Line 193: 35 hail events in 4 years? This would mean an average of >8 hailstorms per year in the location of the observatory, which is much higher, than the long-year observed average – and much higher than global estimates for the general area.
Overall it is not mentioned very clearly, which time period is considered for the instruments (only in the abstract).
How confident is the hail classification of the Parsivel2?
14. Line 195: Inconsistent spelling of boxplot
15. Line 197: Eulerian duration? Here it is important to note that hail duration is determined by both hail area in the storm and the propagation speed of the storm.
16. Figure 7: Please relabel the y-axis with number of minutes. Consider using a log scale so that the frequency of the less frequent classes can still be seen.
17. Line 221: The low percentage of hail events is absolutely to be expected. Hail overall is a rare phenomenon, whereas rain is not.
18. Line 226: This is rather small for hail and most stones should be rather spherical at this size.
19. Line 227 and following: There appears to be an issue with the units, is the "s" missing for "m s$^{-1}$"? In later instances there are also inconsistencies on whether there is a space between "m" and "s" or not.
20. Line 227: "Less than"
21. Line 228ff: How do you unambiguously differentiate between hail and drops in Fig. 8? What is your assumed fall velocity for hail at a corresponding size? Please compare to e.g. Heymsfield et al. (2018).
22. Line 231: "Was found"
23. Line 271: significant implies a statistical test. Please avoid this word, if you do not mean statistical significance.
24. Fig 10: The pink contours are not very well visible. Please consider adding 3 more panels with the shading of the entire events to facilitate the comparison with the hail events.
25. Fig 10-14: Panel labels are missing on all figures (a, b etc)
26. Line 283: Please use a precise time format with the time zone indicator. In addition, this does not match the time shown in Fig. 11 – please convert all time labels to the same time zone. (Fig. 11 vs 12 and 13 vs 14 and correspondingly in the text).
Please also place Figs. 11 and 12 on the same page and then 13 and 14 as well so that the timing of the Parsivel2 classification can be easily compared to the vertical profiles.

27. Line 297f: How does this altitude compare to other studies?
28. Line 308: Fig. 10 implies that between 2 and 4 km, values of 1.5-2 m/s spectral width are rather unusual for rain and more typical for hail. The discussion here seems contradictory. Especially given the statement in Line 293, that values >2.5 m/s are to be considered as outliers anyway.
29. Line 317: The discussion of this event is a bit confusing. First of all, the Parsivel2's classification is also unreliable and if it does not match well with the vertical profiles should also be questioned. Secondly (assuming Parsivel2 as truth): At LT19:25 there is high reflectivity followed by strong attenuation, very high fall velocities exceeding the Nyquist velocity, correspondingly high spectral width and an increased LDR. This does not seem that questionable for hail fall. We also know nothing about the larger spatial structure of the storm. Is the main updraft separated from the principal precipitation areas (I.e. as in supercellular convection)?
30. Line 328: What would the assumed difference be for hail?
31. Line 346: The difficulties of hail observations with a cloud radar could use a more detailed discussion. How is each variable affected by attenuation and resonance scattering?
32. Line 349f: The denominator of the LDR is dominated by ZHH, which is dominated by the largest particles (e.g. Oue et al., 2015). Hence LDR should be mostly governed by the presence of hail. Why not additionally use KDP (e.g. Trömel et al., 2017) and possibly RhoHV as well for hail identification? KDP could help with attenuation issues. (e.g. Kennedy et al., 2001)
For polarimetric hail identification on scanning radars, there is a whole host of additional literature, but of course it does not apply directly to vertical-pointing radars.
33. Line 371: Looking up weather code "SYNOP wawa4860" only leads to this publication. Please include a reference that defines what this is.
34. Lines 372ff: It is unclear which statements refer to the observational record and which refer to the Parsivel2. I.e. Parsivel's annual hail occurrence is approximately 8 per year – this notable difference with the observed frequency should be discussed somewhere.
35. Line 377: Given that this study does not focus on hail stone size and the size is not discussed with respect to the radar signatures, this statement is a far reach. Both events come with different sizes and their different vertical profile might be tied to the hail size. Moreover, two events are not enough to establish relationships with hail size.
36. Line 402: Twice Williams
37. Line 409ff: The outlook is very vague and seems a bit over the top. What kind of applications and insights are you aiming for in future? How can the research benefit the affected communities?

**Suggested References**

- Taszarek, M., Allen, J.T., Marchio, M. *et al.* Global climatology and trends in convective environments from ERA5 and rawinsonde data. *npj Clim Atmos Sci* **4**, 35 (2021). https://doi.org/10.1038/s41612-021-00190-x
- Oue, M., M. R. Kumjian, Y. Lu, J. Verlinde, K. Aydin, and E. E. Clothiaux, 2015: Linear Depolarization Ratios of Columnar Ice Crystals in a Deep Precipitating System over the Arctic Observed by Zenith-Pointing Ka-Band Doppler Radar. *J. Appl. Meteor. Climatol.*, **54**, 1060–1068, https://doi.org/10.1175/JAMC-D-15-0012.1.

- Kennedy, P. C., S. A. Rutledge, W. A. Petersen, and V. N. Bringi, 2001: Polarimetric Radar Observations of Hail Formation. *J. Appl. Meteor. Climatol.*, **40**, 1347–1366, https://doi.org/10.1175/1520-0450(2001)040<1347:PROOHF>2.0.CO;2.
- Andreas F. Prein, Greg J. Holland, Global estimates of damaging hail hazard, Weather and Climate Extremes, Volume 22, 2018, Pages 10-23, ISSN 2212-0947, https://doi.org/10.1016/j.wace.2018.10.004.
- Raupach, T. H., Martius, O., Allen, J. T., Kunz, M., Lasher-Trapp, S., Mohr, S., Rasmussen, K. L., Trapp, R. J., and Zhang, Q.: The effects of climate change on hailstorms, Nature Reviews Earth and Environment, 2, 213–226, https://doi.org/10.1038/s43017-020-00133-9, 2021.
- Sokol, Zbyněk, Jana Minářová, and Petr Novák. 2018. "Classification of Hydrometeors Using Measurements of the Ka-Band Cloud Radar Installed at the Milešovka Mountain (Central Europe)" *Remote Sensing* 10, no. 11: 1674. https://doi.org/10.3390/rs10111674
- Wang F, Liu Y, Zhou Y, Sun R, Duan J, Li Y, Ding Q and Wang H 2023 Retrieving Vertical Cloud Radar Reflectivity from MODIS Cloud Products with CGAN: An Evaluation for Different Cloud Types and Latitudes *Remote Sensing* **15** 816 Online: http://dx.doi.org/10.3390/rs15030816
- Heymsfield, A., M. Szakáll, A. Jost, I. Giammanco, and R. Wright, 2018: A Comprehensive Observational Study of Graupel and Hail Terminal Velocity, Mass Flux, and Kinetic Energy. *J. Atmos. Sci.*, **75**, 3861–3885, https://doi.org/10.1175/JAS-D-18-0035.1.
- Trömel, S., A. V. Ryzhkov, M. Diederich, K. Mühlbauer, S. Kneifel, J. Snyder, and C. Simmer, 2017: Multisensor Characterization of Mammatus. *Mon. Wea. Rev.*, **145**, 235–251, https://doi.org/10.1175/MWR-D-16-0187.1.

---

## Author Comment (AC1)

Overall response:
We appreciate the comments of reviewer 1. Following the reviewers' suggestions, several changes were made to the manuscript. The main changes are:
- A description of the synoptic conditions of the study cases was included.
- Satellite observations (GOES-16) were included to analyze the dynamics of the study cases.
- An analysis of the Doppler spectra of the study cases was included, with special focus on the spectral LDR.
- The discussions and conclusions were rewritten to provide more information on the study results and contextualization.

Reviewer 1:

General comments:
1. The limitations and underlying assumptions for the data and instruments could be discussed more explicitly. It is unconventional to use a cloud radar for a convective study, partially because of attenuation. While this is part of the novelty of this study, this also warrants a discussion. Moreover, there is little to no discussion about the hail detection of Parsivel2 or the reliability of the observer hail reports.
Answer: Now we extended the discussion including Parsivel2 reliability and observer hail reports.

2. There is a lack of connection between the discussion of the long-term hail reports and the microphysical assessment of 2 cases. What is the ulterior motive of diving into these 2 cases in particular? Should the detailed evaluation build the foundation for a later hail-classification algorithm the vertical profiles? This could be a clear motivation in the direction of method development. The two cases alone are also not enough to make a general statement about the overall diversity of hail profiles.
Answer: To address the reviewer's concern, we have expanded our discussion to clarify the intent behind delving into these cases. The detailed evaluation of these instances is intended to serve as a preliminary foundation for the development of a more robust hail-classification algorithm that leverages vertical profile data, we emphasize that these case studies are illustrative examples rather than a comprehensive representation of hail event diversity.
In the revised version of our manuscript, we have enriched our discussion to better articulate how these case studies contribute to the overarching goals of our research. We also underscore the necessity for further analysis encompassing a broader spectrum of hail events to refine and validate the proposed classification algorithm.

3. Both case studies would benefit from a (brief) synoptic discussion of their cases. Is there satellite data available to e.g. estimate convective mode or storm structure? Where did the storms originate and how was their track oriented? How was the synoptic weather situation, are they related to fronts? The overall storm characteristics and atmospheric conditions impact hail development and can help understand the differing evolution of the events.
Answer: We have incorporated an in-depth discussion of the atmospheric conditions surrounding each event, alongside an examination of available satellite data. This enhanced analysis aims to contextualize the convective modes, storm structures, origins, trajectories,

and their relationship with the broader synoptic weather patterns, providing a clearer understanding of the factors influencing hail development in each case.

4. There is a lack of contextualization of the results. A discussion of which findings are in line with previous research and which aspects contradict other studies would be very helpful. Partially the suggestions in the outlook have been implemented in other research (Sokol et al., 2018 -> convection and hail in a Ka-band-radar; Wang et al., 2023 -> hail classification on a satellite borne radar), this at least warrants a mention. Studies focusing on winter precipitation also go into more detail on hydrometeor classification and microphysical understanding with cloud radars and could be used as a comparison. These also often address issues mentioned here (e.g. dealiasing a spectrum).
Answer: Following the reviewer's suggestion now our manuscript explicitly compares and contrasts our findings with existing literature.

Specific comments:
1. Line 44: Space missing before bracket
Answer: Corrected

2. Line 51: Climatological trends depend a great deal on the length of observations and tend to only properly emerge after >40 years. Hence conflicting results are often an artifact of the length of the time series and it would be helpful to include this duration here. Consider including the assessment of Taszarek et al. (2021) on changes in severe convective environments.
Answer: In this study we are including all the historical dataset that we could find. This is a good observation to include as part of our discussion, but we were unable to find the reference of "tend to only properly emerge after >40 years", we would appreciate the reviewer providing the reference for that statement.

3. Line 53: Twice Beal
Answer: Corrected

4. While this is not emphasized in the introduction, convective studies in complex terrain and at high altitudes are rather rare and deserve particular emphasis. How does this compare to other complex terrain studies and hail trends in other mountain regions?
Answer:

5. Section 2: To introduce the region it would be great to have a map of the area. Are the observed hail reports gathered for a larger area than just the observatory? What is considered the representative area for these reports and how does it compare to the area of the Parsivel2?
Answer: We included a map of the study area.

6. Line 128: What is the representative measurement area of a Parsivel2?
Answer: There are no studies about the representativeness of hail observations of Parsivel2 in the Andes, not even for rain gauges, due to the lack of observations in the area. The case study might give a rough idea about the size of the events.

7. Line 142: Consider including the specification Ka-band

Answer: We included Ka-band specification.

8. Line 162: Hail can be very spherical, especially while it is smaller. However, LDR is also influenced by the mix of phases and density in the hailstones.
Answer: We updated the LDR description with a more proper description.

9. Section 3.1: Is there any assessment of how robust the hail reports are? What were the observation criteria? How does the overall annual hail frequency compare to global hail frequency assessments (Prein, Taszarek, Raupach, …)?
Answer: The reports we have are likely for graupel and hail, since the meteorological observer might not take in account the hail size. In this new version we are explicitly discussing this issue.

10. Fig. 1 and following: please define the lines of the boxplot (mean + standard deviation / median + quartiles / …)
Answer: Box plots are described now.

11. Line 185 (and throughout the manuscript): Please consider using a more precise time format that includes the time zone. Moreover, "hours" usually refers to a duration and not a time of day.
Answer: We are specifying now that we are using local time (LT) instead of say "hours".

12. Line 189: twice Tokay
Answer: Corrected

13. Line 193: 35 hail events in 4 years? This would mean an average of >8 hailstorms per year in the location of the observatory, which is much higher, than the long-year observed average – and much higher than global estimates for the general area. Overall it is not mentioned very clearly, which time period is considered for the instruments (only in the abstract). How confident is the hail classification of the Parsivel2?
Answer: We included a more extended discussion about the reliability of Parsivel2 to detect hail. We could not find literature about the reliability of Parsivel2 on classification of hail. Is likely that most of the "hail" are indeed graupel.

14. Line 195: Inconsistent spelling of boxplot
Answer: Corrected.

15. Line 197: Eulerian duration? Here it is important to note that hail duration is determined by both hail area in the storm and the propagation speed of the storm.
Answer: We are measuring with instruments fixed in space, as is described in the methodology. The satellite observation for the case studies can give an idea about the lifetime of the event but not the hail area.

16. Figure 7: Please relabel the y-axis with number of minutes. Consider using a log scale so that the frequency of the less frequent classes can still be seen.
Answer:

17. Line 221: The low percentage of hail events is absolutely to be expected. Hail overall is a rare phenomenon, whereas rain is not.

Answer: We rewrite this sentence to reduce the emphasis on this obvious result.

18. Line 226: This is rather small for hail and most stones should be rather spherical at this size.

Answer: The hailstones in the Andes might be significantly smaller than other areas.

19. Line 227 and following: There appears to be an issue with the units, is the "s" missing for "m s-1"? In later instances there are also inconsistencies on whether there is a space between "m" and "s" or not.

Answer: Corrected.

20. Line 227: "Less than"

Answer: Corrected.

21. Line 228ff: How do you unambiguously differentiate between hail and drops in Fig. 8? What is your assumed fall velocity for hail at a corresponding size? Please compare to e.g. Heymsfield et al. (2018).

Answer: We are using the previously analyzed (Valdivia 2020) raindrop velocities to diagnose the presence of raindrops (The rain drops curves are displayed in the Figure).

22. Line 231: "Was found"

Answer: Corrected.

23. Line 271: significant implies a statistical test. Please avoid this word, if you do not mean statistical significance.

Answer: Corrected.

24. Fig 10: The pink contours are not very well visible. Please consider adding 3 more panels with the shading of the entire events to facilitate the comparison with the hail events.

Answer: We updated the figure with 3 more panels as it was suggested.

25. Fig 10-14: Panel labels are missing on all figures (a, b etc)

Answer: Corrected.

26. Line 283: Please use a precise time format with the time zone indicator. In addition, this does not match the time shown in Fig. 11 – please convert all time labels to the same time zone. (Fig. 11 vs 12 and 13 vs 14 and correspondingly in the text). Please also place Figs. 11 and 12 on the same page and then 13 and 14 as well so that the timing of the Parsivel2 classification can be easily compared to the vertical profiles.

Answer: Corrected. We are now using the local time as reference in all the figures.

27. Line 297f: How does this altitude compare to other studies?

Answer:

28. Line 308: Fig. 10 implies that between 2 and 4 km, values of 1.5-2 m/s spectral width are rather unusual for rain and more typical for hail. The discussion here seems contradictory.

Especially given the statement in Line 293, that values >2.5 m/s are to be considered as outliers anyway.

Answer: We did not find any typical range of spectral for hail, this variable is usually greater in presence of turbulence, which can be related to hail production but that specific case shows a steady profile near the surface.

29. Line 317: The discussion of this event is a bit confusing. First of all, the Parsivel2's classification is also unreliable and if it does not match well with the vertical profiles should also be questioned. Secondly (assuming Parsivel2 as truth): At LT19:25 there is high reflectivity followed by strong attenuation, very high fall velocities exceeding the Nyquist velocity, correspondingly high spectral width and an increased LDR. This does not seem that questionable for hail fall. We also know nothing about the larger spatial structure of the storm. Is the main updraft separated from the principal precipitation areas (I.e. as in supercellular convection)?

Answer: This is a good point. We removed this sentence, and included a more detailed analysis of LDR and spectral LDR.

30. Line 328: What would the assumed difference be for hail?

Answer: We meant that the differences are hardly noticeable from a only raindrop vel vs diameter relationship.

31. Line 346: The difficulties of hail observations with a cloud radar could use a more detailed discussion. How is each variable affected by attenuation and resonance scattering?

Answer: We extended the discussion including the new results.

32. Line 349f: The denominator of the LDR is dominated by $Z_{HH}$, which is dominated by the largest particles (e.g. Oue et al., 2015). Hence LDR should be mostly governed by the presence of hail. Why not additionally use $K_{DP}$ (e.g. Trömel et al., 2017) and possibly RhoHV as well for hail identification? $K_{DP}$ could help with attenuation issues. (e.g. Kennedy et al., 2001) For polarimetric hail identification on scanning radars, there is a whole host of additional literature, but of course it does not apply directly to vertical-pointing radars.

Answer: Our radar works at single polarization so the only polarimetric variable that we have is LDR since we are working at vertical incidence.

33. Line 371: Looking up the weather code "SYNOP wawa4860" only leads to this publication. Please include a reference that defines what this is.

Answer: The term "SYNOP wawa" adheres to a standardized coding system as per the guidelines of the World Meteorological Organization (WMO, 2019), specifically within the FM system of numbering code forms for reporting weather conditions. In the WMO Manual on Codes (WMO-No. 306), the "SYNOP" code (FM 12 numeric system) refers to the reporting of surface observations from a fixed land station. In specific the wawa stands for automatic surface observations. In our manuscript, the "SYNOP wawa 4860" code is utilized to categorize specific meteorological conditions captured by the OTT Parsivel2 disdrometer, as detailed in its Operating Instructions Manual. We have now included references to both the WMO Manual on Codes and the OTT Parsivel2 Operating Instructions Manual in the revised manuscript.

World Meteorological Organization (2019). Manual on Codes (WMO-No. 306), Volume I.1, Part A – Alphanumeric Codes. Retrieved from: https://library.wmo.int/doc_num.php?explnum_id=10235 (Table 4680, p. A-360)
OTT (2016). Operating Instructions Manual: OTT Parsivel2. Retrieved from: https://www.ott.com/download/operating-instructions-present-weather-sensor-ott-parsivel2-without-screen-heating-1/ (Appendix D, Table 4680).

34. Lines 372ff: It is unclear which statements refer to the observational record and which refer to the Parsivel2. I.e. Parsivel's annual hail occurrence is approximately 8 per year – this notable difference with the observed frequency should be discussed somewhere.
Answer: We included this observation in the discussion section.

35. Line 377: Given that this study does not focus on hail stone size and the size is not discussed with respect to the radar signatures, this statement is a far reach. Both events come with different sizes and their different vertical profile might be tied to the hail size. Moreover, two events are not enough to establish relationships with hail size.
Answer: We remove that sentence.

36. Line 402: Twice Williams
Answer: Corrected.

37. Line 409ff: The outlook is very vague and seems a bit over the top. What kind of applications and insights are you aiming for in future? How can the research benefit the affected communities?
Answer: We remove this vague conclusion.

---

## Author Comment (AC2)

Reviewer 2:

This study analyzed hail reports as well as disdrometer and upward pointing radar measurements of hail events at one station in the central Andes. The text, including structure, title, and abstract, and the Figures are of good quality. In my opinion, given my specific comments 2-4, the results lack representativeness and are not very impactful. However, some of the results are perhaps worth publishing and given the wealth of different analyses, including recent topics such as polarimetric radar identification, I could see the study having some relevance for the community. Some points should be clarified before publication.

Overall response:
We appreciate the comments of reviewer 2. Following the reviewers' suggestions, several changes were made to the manuscript. The main changes are:
- A description of the synoptic conditions of the study cases was included.
- Satellite observations (GOES-16) were included to analyze the dynamics of the study cases.
- An analysis of the Doppler spectra of the study cases was included, with special focus on the spectral LDR.
- The discussions and conclusions were rewritten to provide more information on the study results and contextualization.

Specific comments:

1.  I. 97 a map with elevation and the station location would be very helpful I think

    Answer: We included a map for the study area.

2.  I. 97 Hail is typically infrequent on high mountain regions and more frequent in foothills (see Allen 2017 for Rocky mountains in US and Punge 2017 for the European Alps). I'm missing some justification for why this location was investigated. Is hail an often observed threat there (this doesn't seem to be the case from the hail sizes that you observed, <14mm)? If not, I think it should be made even clearer (in intro and summary especially) that the observations are only representative for the high mountain range, not other regions of Peru (probably not even the whole mantaro river valley which you mention in the intro, give its complex terrain).

    Answer: This the only hail observation so far in the central Andes. The meteorological observations overall are very scarce in the Andes.The Huancayo Observatory is trying to fill this gap. Also, the hail is a problem for agriculture in the zone as is explained in the introduction.

3.  I. 97-103 Are there any other observatories such as this one in Peru which you could add to the analysis? This would increase the robustness of the results.
    Answer: We are using all the data set that we were able to find, there are not any other reports nor study about hail observations in the central Andes.

4. I'd also personally be very interested in the spatial distribution as our satellite-based detections show some strong activity in Peru, but more in the foothills to the East. Perhaps you can share your experience on this.

   Answer: I included satellite observations from GOES16 for the study cases. It seems like the strong activity occurs only at a small scale in the Andes.

5. section 2 in general: Do you have events of Graupel in this region? How did you make sure events were correctly classified as hail? Was there a minimum size? Later you comment on the observed sizes (line 340), which are almost exclusively small, so I'd wager some of the cases were in fact Graupel. Graupel is also not considered in the Disdrometer (probably because it is indistinguishable from small hail for there), which leads me to suspect that many of the "Hailstorms" in section 3.2 are Graupel. I think this is very important because Graupel can never become large, while hail can. If most of your events are Graupel, then calling this study an analysis of hail events is very misleading. This should at least be made very clear already in the intro and methods, not late in line 340.

   Answer:  This is a good point, actually we cannot differentiate between hail and graupel in this study. The Parsivel does not make the distinction and the hail reports are likely including graupel as well. The issue is not stated in the manuscript.

6. l. 297 (also l. 385) I'm not a microphysics person but in the reference below, high LDR aloft is attributed to the alignment of ice chips in the electric fields (Melnikov, et al. 2019).
   Perhaps in general, you could compare your results more to those of other studies (I'm sure the one I found is not the only one using polarimetric radar signatures to classify hail)

   Answer: We now rewrite the LDR description to accuracy.

7. l. 315-319 The radar only sees a vertical section at one point, not the whole 3D cloud, while hail growth happens in 3D trajectories and the stones are advected relative to the updraft (even several km outside the main updraft in some cases), see e.g. Kunjian et al. 2020. Perhaps some perspective on how you relate your 1D (or 2D segments) of measurements to these 3D processes would be insightful. It wasn't clear to me what you mean by "Intuition".

   Answer: In fact now we included satellite observation we realized the radar is only observing a small part of the storm. The issue is now detailed in the manuscript. We rewrote the sentence that included "Intuition".

Technical corrections and suggestions:

1. l. 44 add space after "severity"
   Answer: Corrected.

2. l. 47 I suggest adding Allen et al. 2018 as reference in the brackets
   Answer: Thank you, we are now citing Allen.
3. l. 52 remove "Beal" once
   Answer: Corrected.
4. l 60-79 repeating all these points here seems a bit redundant. Consider summarizing them in 2-3 sentences or picking the ones most relevant for your study (such as the microphysical aspects)
   Answer: Actually, we are making a resume of the Raupach recommendations, we think that give a good insight of the state of art, for our study.
5. l 80-81 "In the Andes" is repeated 3 times here
   Answer: Corrected.
6. l. 101 The text seems broken here, do you mean "...sorrounding the observatory."?
   Answer: We removed this broken text.
7. l. 106-107 I don't understand this sentence. Weren't the reports localized before 2016 as well?
   Answer: The reports were discontinued in early 2016. So our 2016 hail events is 0. I specified this issue in the text now.
8. l. 107-110 I recommend starting this sentence with "However, instruments ..."
   Answer: Suggestion accepted.
9. sections 2.1 and 2.2 A lot of this technical information seems irrelevant for the study. I recommend sticking to what is important for hail accuracy and just include a reference for further reading
   Answer:
10. Fig. 3 and elsewhere: I'm assuming the times are in local time not UTC? Be clear (if I haven't overread it)
    Answer: Now we are being specific with the local time use.
11. section 3.3: Just a question, do you think the LDR relationship typically observed in horizontal pointing radar (higher LDR for hail) is better or worse here because you use a verticall pointing radar, which sees falling raindrops from the bottom?
    Answer: Certainly it is easier to observe microphysical properties at vertical incidence due to high vertical (and time) resolution, Also the velocity is a good proxy of the particle sizes, information that is more complex to derive at horizontal incidence.
12. l.334 "Friedrich" twice
    Answer: Corrected.
13. paragraph starting at l. 328 Good discussion of weaknesses!
    Answer: Thanks, the reliability of Parsivel2 to identify hail is still under study, which is part of this work.
14. l. 405-413 Just my opinion, but his high self-appraisal seems a bit unprofessional. This is an interesting "pioneering study" yes, but I recommend being a bit more humble about the impact in a scientific publication, if it is not ground breaking.
    Answer: You are totally right, we rewrite the conclusion section. Now we included the new results as part of the conclusions.

---

## Author Response (AR2)

**Based on the second round of review, the manuscript has been considered suitable for publication subject to the following few technical correction:**

line 25: Here it could be mentioned that only ice pellets >5mm are considered hail
Answer: We included the phrase "Hail is defined as ice pellets that exceed 5 mm in diameter."

line 26: in the end of the sentence you could mention that <2cm can already damage crops (since this is significant to your region)
Answer: We included the phrase "However, even hailstones smaller than 2 cm can cause significant damage to crops." at the end of the paragraph.

line 220: "have been discarded"
Answer: Corrected.

Fig. 14: the caption doesn't mention the red circle in the plots
Answer: We removed the circles, since they are not relevant in this study.

lines 396-399: Somewhere here it should be mentioned that the trend (-0.5) is based on observations at ONE location and hence might not be robust
Answer: We rewrite this phrase "The analysis showed a decrease in the frequency of hailstorms over time (-0.5/decade)," to "The analysis showed a decrease in the frequency of hailstorms over time, at a rate of -0.5 per decade, and this finding is based on observations from a single location."

lines 484 and 486: upper case "-1"
Answer: Corrected.